# Neural QAOA²: Differentiable Joint Graph Partitioning and Parameter Initialization for Quantum Combinatorial Optimization

**Zubin Zheng** [* 1]  **Jiahao Wu** [* 1]  **Shengcai Liu** [1]

## Abstract

The quantum approximate optimization algorithm (QAOA) holds promise for combinatorial optimization but is constrained by limited qubits. While divide-and-conquer frameworks like QAOA² address scalability by partitioning graphs into subgraphs, existing methods suffer from two fundamental limitations: i) misalignment between heuristic partitioning metrics and quantum optimization goals, and ii) topology-blind parameter initialization that leads to optimization cold starts. To bridge these gaps, we propose **Neural QAOA²**, an end-to-end differentiable framework that jointly generates graph partitions and initial parameters. By integrating a generative evaluative network (GEN), our method utilizes a differentiable quantum evaluator as a high-fidelity performance surrogate to provide direct gradient guidance, enabling the joint generator to learn the intrinsic mapping from graph topology to high-quality partition and parameter configurations. Extensive experiments on 183 QUBO, Ising, and MaxCut instances (21 to 1000 variables) demonstrate that our gradient-driven approach broadly outperforms heuristic baselines, ranking first on 101 instances. It exhibits zero-shot generalization across out-of-distribution graph topologies and scales.

## 1. Introduction

Quantum computing offers a transformative approach to NP-hard combinatorial optimization (Gemeinhardt et al., 2023). The quantum approximate optimization algorithm (QAOA, Farhi et al., 2014) has emerged as a prominent approach

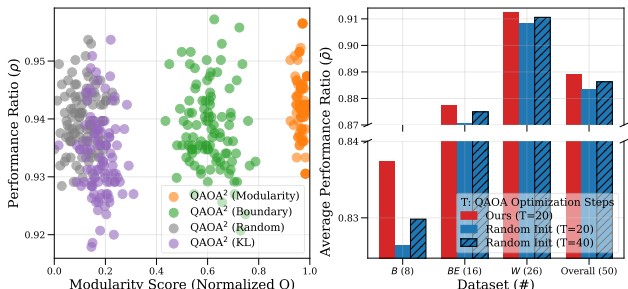

*Figure 1.* **Existing D&C frameworks suffer from two fundamental limitations.** Left: misalignment between partitioning metrics and optimization goals. Graph-theoretic metrics (e.g., modularity) fail to align with quantum solution quality, showing negligible correlation (Pearson's $r = 0.2859$ on instance *g05_100.1*) with the performance ratio. Right: topology-blind parameter initialization. Random initialization suffers from cold starts; even with double optimization steps ($T = 40$), it fails to match our topology-aware initialization ($T = 20$) across all datasets.

for quadratic unconstrained binary optimization (QUBO) problems (Lucas, 2014). While QAOA holds the potential for quantum advantage, a critical bottleneck remains: real-world applications typically involve thousands of variables, whereas current quantum resources are limited to the hundred-qubit scale (Preskill, 2018; Harrigan et al., 2021; Kim et al., 2023; Google Quantum AI and Collaborators, 2025). Consequently, enhancing algorithmic scalability is pivotal for realizing "quantum utility" (Kim et al., 2023).

To address scalability, the divide-and-conquer (D&C) strategy has become widely adopted (Guerreschi, 2021; Li et al., 2022; Zhou et al., 2023). Within this D&C paradigm, QAOA-in-QAOA (QAOA², Zhou et al., 2023) has emerged as a representative framework. It tackles MaxCut (equivalent to QUBO) by solving partitioned subgraphs via QAOA, and then merging these sub-solutions to derive the global solution. By leveraging the $\mathbb{Z}_2$ symmetry to reformulate the merging step, QAOA² enables large-scale instances to be processed on small-scale quantum devices.

However, existing D&C frameworks suffer from two fundamental limitations: **i) Misalignment between partitioning metrics and optimization goals.** Existing frameworks rely on handcrafted partitioning heuristics (e.g., modularity,

*Equal contribution  [1]Guangdong Provincial Key Laboratory of Brain-Inspired Intelligent Computation, Department of CSE, SUSTech. Correspondence to: Shengcai Liu <liusc3@sustech.edu.cn>.

*Proceedings of the 43rd International Conference on Machine Learning*, Seoul, South Korea. PMLR 306, 2026. Copyright 2026 by the author(s).

Clauset et al., 2004) that optimize generic graph-theoretic metrics. However, these metrics are decoupled from the optimization goal. As a result, optimizing such topological proxies often fails to translate into high-quality quantum solutions. This is empirically confirmed in Figure 1 (Left), which reveals a negligible correlation between these metrics and global solution quality. **ii) Topology-blind parameter initialization.** When solving subgraphs, existing frameworks employ random initialization for quantum circuit parameters $(\gamma, \beta)$, blind to the intrinsic mapping between parameters and graph topology (Katial et al., 2025). This unguided initialization leads to an optimization cold start (Jain et al., 2022; Liang et al., 2024). As evidenced in Figure 1 (Right), even doubling the optimization steps with random initialization fails to match a topology-aware strategy, imposing substantial iteration overhead.

To bridge these gaps, we propose **Neural QAOA²**, which reimagines the D&C paradigm by integrating a differentiable generative evaluative network (GEN) composed of a quantum evaluator and a joint generator. To resolve the metric misalignment (limitation i), the quantum evaluator functions as a high-fidelity surrogate, offering direct gradient guidance to ensure partitions are optimized for quantum performance rather than heuristic proxies. In parallel, addressing the initialization bottleneck (limitation ii), the joint generator exploits this guidance to learn a topology-aware mapping, directly synthesizing high-quality circuit parameters $(\gamma, \beta)$ conditioned on the partitions. Crucially, to enable gradient flow through discrete structures, we introduce a differentiable discretization mechanism, ensuring that this joint learning process remains fully end-to-end trainable. Finally, our framework adopts a hybrid strategy that captures generalizable patterns through offline pre-training and performs instance-specific refinement via test-time adaptation.

The main contributions of this work are summarized below.

- We propose Neural QAOA², a novel framework pioneering gradient-driven scalable quantum combinatorial optimization. By integrating a quantum evaluator and a joint generator into a unified generative evaluative network, our method effectively resolves the metric misalignment and topology-blind initialization limitations.

- We devise specialized mechanisms to unlock the trainability of the discrete partitioning: an orthogonal complement head that imposes a geometric inductive bias to construct discriminative soft partitions, and a greedy capacity discretization with straight-through estimator to enable gradient backpropagation while strictly satisfying qubit constraints.

- Extensive experiments on 183 QUBO, Ising, and MaxCut instances (21 to 1000 variables) show that Neural QAOA² broadly outperforms heuristic baselines. It ranks first on 101 out of 183 instances, nearly doubling the 53 wins of

the strongest runner-up, and demonstrates strong generalization across both in-distribution and out-of-distribution graph topologies and scales.

## 2. Related Work

**Scalable QAOA via D&C.** D&C approaches for scalable QAOA include QAOA² (Zhou et al., 2023) and DC-QAOA (Li et al., 2022). Among them, QAOA² is a representative D&C approach. However, this method relies on heuristic proxy metrics, such as modularity (Clauset et al., 2004), boundary size (Guerreschi, 2021), and cut size (Kernighan & Lin, 1970), which are not directly aligned with the ultimate quantum optimization objective. Building upon the D&C framework, Neural QAOA² addresses this by introducing a differentiable quantum evaluator to predict quantum performance, providing gradients to guide partition generation.

**Parameter Initialization in QAOA.** Initialization strategies are generally categorized into topology-blind (e.g., random), physics-inspired (e.g., TQA, Sack & Serbyn, 2021; INTERP/FOURIER, Zhou et al., 2020), and learning-based approaches (e.g., QIBPI, Katial et al., 2025). However, integrating these strategies into D&C frameworks yields suboptimal results (see detailed analysis in Appendix C.1). Decoupled from partitioning, they fail to adapt to specific subgraph structures. Notably, standard topology-blind strategies ignore the intrinsic topology-parameter mapping, causing optimization cold starts (Jain et al., 2022). In contrast, our joint generator explicitly captures this mapping, enabling an efficient warm start tailored to the partition.

## 3. Preliminaries

**Equivalence between QUBO and MaxCut.** QUBO minimizes the objective $f(\boldsymbol{x}) = \sum_{i,j} Q_{ij} x_i x_j + \sum_i c_i x_i$ for binary variables $\boldsymbol{x} \in \{0, 1\}^n$. This problem is mathematically equivalent to finding the ground state of an Ising Hamiltonian. By substituting $x_i = (1 - z_i)/2$ and absorbing linear terms via auxiliary spins (Glover et al., 2022; Lucas, 2014), QUBO formally reduces to weighted MaxCut on a graph $G = (V, E)$, where $|V| = N = n + 1$. MaxCut aims to find a spin configuration $\boldsymbol{z} \in \{+1, -1\}^N$ that maximizes the Hamiltonian $H_C = \frac{1}{2} \sum_{(u,v) \in E} w_{uv}(1 - z_u z_v)$, where $w_{uv}$ denotes edge weight. In the quantum setting, classical variables $z_u$ are promoted to Pauli-$Z$ operators $\hat{\sigma}_u^z$ to construct the quantum Hamiltonian $\hat{H}_C$.

**QAOA.** This variational algorithm (Farhi et al., 2014) aims to approximate the ground state of $\hat{H}_C$, which corresponds to the optimal MaxCut solution. It employs a parameterized

ansatz generated by a depth-$p$ quantum circuit:

$$|\psi(\boldsymbol{\gamma}, \boldsymbol{\beta})\rangle = \prod_{l=1}^{p} U_B(\beta_l) U_C(\gamma_l)|+\rangle^{\otimes N}, \quad (1)$$

where $|+\rangle^{\otimes N}$ is the uniform superposition state of $N$ qubits, and $(\boldsymbol{\gamma}, \boldsymbol{\beta}) \in [0, 2\pi]^{2p}$ are variational parameters. The problem unitary $U_C(\gamma_l) = e^{-i\gamma_l \hat{H}_C}$ encodes the cost function, while the mixer unitary $U_B(\beta_l) = e^{-i\beta_l \sum_{j=1}^{N} \hat{\sigma}_j^x}$ drives transitions using Pauli-$X$ operators $\hat{\sigma}_j^x$. The parameters are optimized to maximize the expectation value:

$$(\boldsymbol{\gamma}^*, \boldsymbol{\beta}^*) = \arg\max_{\boldsymbol{\gamma}, \boldsymbol{\beta}} \langle\psi(\boldsymbol{\gamma}, \boldsymbol{\beta})|\hat{H}_C|\psi(\boldsymbol{\gamma}, \boldsymbol{\beta})\rangle. \quad (2)$$

Upon convergence, measuring the state $|\psi(\boldsymbol{\gamma}^*, \boldsymbol{\beta}^*)\rangle$ yields high-quality solution bitstrings.

**QAOA².** Standard QAOA requires $N$ qubits to solve an $N$-node graph, limiting scalability. To overcome this, QAOA² (Zhou et al., 2023) employs a D&C framework leveraging $\mathbb{Z}_2$ symmetry. It first partitions the graph $G$ into subgraphs $\{G_i\}_{i=1}^{k}$ fitting available quantum resources, and independently optimizes local solutions $\boldsymbol{z}^{(i)}$ via QAOA. To reconstruct the global solution, the framework determines an optimal polarity $s_i \in \{+1, -1\}$ (flip or not) for each local solution. This merging process is strictly equivalent to solving a new MaxCut problem on a reduced graph with $k$ nodes: $\max_{\boldsymbol{s}} \frac{1}{2} \sum_{i<j} w'_{ij}(1 - s_i s_j)$, where the weight $w'_{ij}$ quantifies the net cut gain between subgraphs conditioned on their local solutions. QAOA is then applied to solve this reduced problem to merge local solutions according to $\boldsymbol{s}$. If the reduced problem size still exceeds qubit limits, it is recursively partitioned, solved and reformulated until it fits the available number of qubits. Finally, the global solution is recovered by propagating optimal polarities top-down to merge low-level local solutions.

## 4. Neural QAOA²

We propose Neural QAOA², a unified divide-and-conquer framework that reimagines the paradigm by integrating a generative evaluative network (GEN). GEN replaces heuristic partitioning and random parameter initialization with a single neural backbone that jointly generates graph partitions and QAOA initial parameters. This end-to-end design aligns partitioning with quantum optimization objectives and enables topology-aware warm starts.

### 4.1. Overview of GEN

The objective of GEN is to identify the graph partition $\mathbf{S}$ and variational parameters $\mathbf{P}$ that maximize the performance ratio $\rho$ of QAOA² on a graph $G = (V, E)$. To resolve the unboundedness and numerical instability arising from

negative edge weights, we define the performance ratio as:

$$\rho = \frac{\text{Cut}(G) - \text{Neg}(G)}{\text{OPT}(G) - \text{Neg}(G)}, \quad (3)$$

where $\text{Cut}(G)$ denotes the cut value under the configuration $(\mathbf{S}, \mathbf{P})$, $\text{OPT}(G)$ is the best-known cut, and $\text{Neg}(G) \leq 0$ is the sum of all negative edge weights. This provides the following guarantee, yielding stable gradient scales.

**Proposition 4.1** (Boundedness). *For any graph $G$, the metric $\rho$, evaluated on the output of the QAOA², is theoretically bounded in $[0.5, 1.0]$ (proof provided in Appendix D).*

To optimize this metric within quantum resource constraints, we formally define the discrete partition matrix $\mathbf{S} \in \{0, 1\}^{N \times k}$ and the continuous parameter matrix $\mathbf{P} \in [0, 2\pi]^{2p \times k}$. Specifically, $\mathbf{S}$ assigns $N$ nodes to $k$ subgraphs subject to the assignment constraint $\sum_{j=1}^{k} \mathbf{S}_{ij} = 1$ (ensuring unique membership) and the capacity constraint $\sum_{i=1}^{N} \mathbf{S}_{ij} \leq \text{max\_nodes}$ (respecting qubit limits). Correspondingly, $\mathbf{P}$ represents the $2p$ variational parameters $(\boldsymbol{\gamma}, \boldsymbol{\beta}) \in [0, 2\pi]^{2p}$ for each of the $k$ subgraphs.

As illustrated in Figure 2, the GEN architecture comprises two components. The quantum evaluator ($f_\phi$) serves as a differentiable surrogate to circumvent expensive quantum circuit simulations and provide gradient guidance for the generator. For a given graph and candidate configuration $(G, \mathbf{S}, \mathbf{P})$, it predicts the performance ratio: $\hat{\rho} = f_\phi(G, \mathbf{S}, \mathbf{P})$. Complementarily, the joint generator ($g_\theta$) is a parameterized generative model that directly infers the discrete partition $\mathbf{S}$ and continuous parameters $\mathbf{P}$ from the graph $G$: $(\mathbf{S}, \mathbf{P}) = g_\theta(G)$. To enable efficient training and inference, we design an optimization protocol consisting of distribution-level learning and instance-level optimization.

**Training Phase: Distribution-level Learning.** In this phase, we aim to learn a generalizable mapping from graph structure distributions to high-quality configurations using an offline dataset $\mathcal{D}_{\text{offline}}$ and its constituent graph instances $\mathcal{D}_{\text{graph}}$. The process involves two sequential stages. First, we train the quantum evaluator $f_\phi$ via supervised learning on labeled data $\mathcal{D}_{\text{offline}} = \{(G_i, \mathbf{S}_i, \mathbf{P}_i, \rho_i)\}$ to minimize the prediction error $\mathcal{L}_{\text{MSE}}$, establishing a high-fidelity surrogate for QAOA² simulations:

$$\min_{\phi} \mathbb{E}_{(G, \mathbf{S}, \mathbf{P}, \rho) \sim \mathcal{D}_{\text{offline}}} [\mathcal{L}_{\text{MSE}}(f_\phi(G, \mathbf{S}, \mathbf{P}), \rho)]. \quad (4)$$

Upon convergence, $f_\phi$ is frozen to serve as a differentiable objective providing gradients $\nabla_\theta f_\phi$. Subsequently, the joint generator $g_\theta$ is trained in an unsupervised manner to maximize the expected performance ratio on $\mathcal{D}_{\text{graph}}$ via gradient ascent guided by $f_\phi$:

$$\max_{\theta} \mathbb{E}_{G \sim \mathcal{D}_{\text{graph}}} [f_\phi(G, g_\theta(G))]. \quad (5)$$

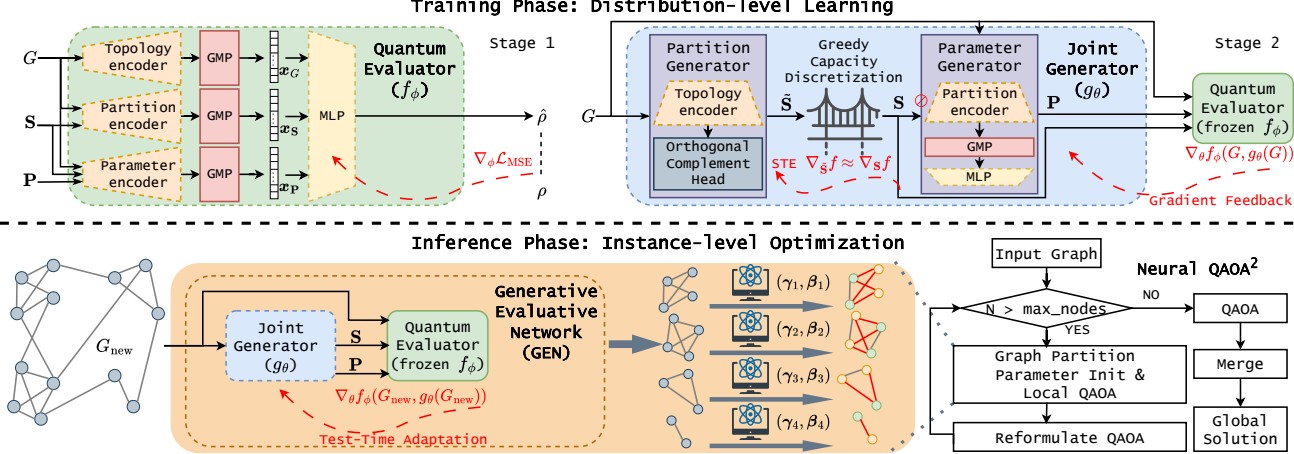

*Figure 2.* **Overview of Generative Evaluative Network.** As the core engine of Neural QAOA², GEN unifies a joint generator ($g_\theta$) and a quantum evaluator ($f_\phi$). $g_\theta$ employs an orthogonal complement head and greedy capacity discretization (visualized as a suspension bridge) to strictly enforce qubit constraints while enabling gradient backpropagation via the straight-through estimator (STE, red dashed line). $f_\phi$ serves as a differentiable surrogate guiding both distribution-level pre-training and instance-level test-time adaptation.

**Inference Phase: Instance-level Optimization.** For an unseen instance $G_{\text{new}}$, we employ a test-time adaptation (TTA) strategy to refine the configuration. Initially, a single forward pass with the pre-trained $g_\theta$ yields a high-quality initialization $(\mathbf{S}_0, \mathbf{P}_0) = g_\theta(G_{\text{new}})$, leveraging the learned prior. To further boost performance, we utilize the frozen $f_\phi$ as an instance-specific objective. We fine-tune the generator parameters $\theta$ via limited-step gradient ascent to obtain:

$$\theta^* = \arg\max_\theta f_\phi(G_{\text{new}}, g_\theta(G_{\text{new}})).\tag{6}$$

The output $(\mathbf{S}^*, \mathbf{P}^*) = g_{\theta^*}(G_{\text{new}})$ transitions from a general distribution prior to an instance-specialized configuration.

### 4.2. Quantum Evaluator

The quantum evaluator $f_\phi(G, \mathbf{S}, \mathbf{P})$ serves as a differentiable surrogate to circumvent expensive physical simulations of QAOA² and provide gradient guidance for the upstream generator. As illustrated in Figure 2, to encode heterogeneous inputs (discrete topologies, binary partitions, and continuous parameters), the evaluator employs a multi-view graph neural network architecture. This design independently processes distinct modalities through three parallel pipelines to map these inputs into a unified latent space:

**Global Topology View.** To capture the global context of graph $G$, the adjacency matrix $\mathbf{A} \in \mathbb{R}^{N \times N}$ and initial node features $\mathbf{X} \in \mathbb{R}^{N \times d}$ are utilized (see Appendix A.3). A topology encoder extracts the global $h$-dimensional embedding $\mathbf{H}_{\text{topology}} = \text{Encoder}_{\text{topology}}(\mathbf{X}, \mathbf{A}) \in \mathbb{R}^{N \times h}$.

**Partition-Induced Subgraph View.** The partition matrix $\mathbf{S}$ dictates edge retention within subgraphs. We dynamically construct a subgraph adjacency matrix $\mathbf{A}_{\text{sub}}$ by masking cross-partition edges:

$$\mathbf{A}_{\text{sub}} = \mathbf{A} \odot (\mathbf{S}\mathbf{S}^\top),\tag{7}$$

where $\odot$ denotes the Hadamard product. Here, $(\mathbf{S}\mathbf{S}^\top)_{ij}$ equals 1 if and only if nodes $i, j$ share the same partition, effectively preserving only intra-subgraph structures. A partition encoder then extracts local features: $\mathbf{H}_{\text{partition}} = \text{Encoder}_{\text{partition}}(\mathbf{X}, \mathbf{A}_{\text{sub}}) \in \mathbb{R}^{N \times h}$.

**Quantum Parameter View.** Parameters $\mathbf{P} \in [0, 2\pi]^{2p \times k}$ are broadcast to the node level via $\mathbf{X}_{\text{param}} = \mathbf{S}\mathbf{P}^\top \in [0, 2\pi]^{N \times 2p}$. To respect the $2\pi$-periodicity of QAOA parameters, we apply sine-cosine embeddings: $\tilde{\mathbf{X}}_{\text{param}} = \text{concat}[\sin(\mathbf{X}_{\text{param}}), \cos(\mathbf{X}_{\text{param}})] \in [-1, 1]^{N \times 4p}$. These features are processed by a parameter encoder operating on $\mathbf{A}_{\text{sub}}$, as parameters function strictly within subgraphs: $\mathbf{H}_{\text{param}} = \text{Encoder}_{\text{param}}(\tilde{\mathbf{X}}_{\text{param}}, \mathbf{A}_{\text{sub}}) \in \mathbb{R}^{N \times h}$.

To predict the performance ratio $\hat{\rho}$, we aggregate the learned representations via global mean pooling (GMP) and concatenation: $\mathbf{H} = \text{concat}[\text{GMP}(\mathbf{H}_{\text{topology}}), \text{GMP}(\mathbf{H}_{\text{partition}}), \text{GMP}(\mathbf{H}_{\text{param}})] \in \mathbb{R}^{3h}$. Finally, an MLP maps $\mathbf{H}$ to the output. To enforce the theoretical bounds derived in Proposition 4.1, we scale the sigmoid activation: $\hat{\rho} = 0.5 \cdot (\text{sigmoid}(\text{MLP}(\mathbf{H})) + 1)$. This ensures the predicted ratio strictly falls within $[0.5, 1.0]$.

### 4.3. Joint Generator

The joint generator $g_\theta(G)$ aims to generate a discrete partition matrix $\mathbf{S}$ constrained by qubit capacity, and the corresponding variational parameters $\mathbf{P}$. We formulate this generation process via the probabilistic chain rule:

$$P(\mathbf{S}, \mathbf{P}|G) = P(\mathbf{S}|G) \cdot P(\mathbf{P}|\mathbf{S}, G).\tag{8}$$

This factorization justifies our sequential "partition-first, parameter-second" architecture: The partition $\mathbf{S}$ determines the local Hamiltonians, serving as the prerequisite for optimizing parameters $\mathbf{P}$. As illustrated in Figure 2, the generator comprises three differentiable stages:

**Soft Partition via OCH.** To model $P(\mathbf{S}|G)$, we first extract node embeddings via a topology encoder: $\mathbf{H}_{\text{topology}} = \text{Encoder}_{\text{topology}}(\mathbf{X}, \mathbf{A}) \in \mathbb{R}^{N \times h}$. Standard GNN training often results in indistinguishable soft probabilities due to embedding over-smoothing (Li et al., 2018) and objective degeneracy (Ying et al., 2018; Bianchi et al., 2020; Tsitsulin et al., 2023), hindering effective training. To address this and construct discriminative soft partitions, we propose the orthogonal complement head (OCH). OCH imposes a strong geometric inductive bias by enforcing the cluster centers $\mathbf{C} \in \mathbb{R}^{k \times h}$ to be mutually orthogonal and orthogonal to the global graph embedding $\boldsymbol{g} = \text{GMP}(\mathbf{H}_{\text{topology}}) \in \mathbb{R}^h$:

$$\mathbf{C}\boldsymbol{g} = \mathbf{0} \quad \text{and} \quad \mathbf{C}\mathbf{C}^\top = \mathbf{I}. \tag{9}$$

Specifically, $\mathbf{C}$ is dynamically constructed by orthogonalizing a randomly initialized matrix with respect to $\boldsymbol{g}$ (e.g., via QR decomposition, Golub & Van Loan, 2013). By constraining centers to the orthogonal complement of the global context, OCH maximizes inter-cluster separability in the embedding space. The soft partition $\tilde{\mathbf{S}}$ is then derived via softmax similarity: $\tilde{\mathbf{S}} = \text{softmax}(\mathbf{H}_{\text{topology}}\mathbf{C}^\top) \in [0, 1]^{N \times k}$. Implementation details of OCH are provided in Appendix A.1.

**Differentiable Discretization.** Converting soft probabilities $\tilde{\mathbf{S}}$ into a binary partition $\mathbf{S}$ requires strictly satisfying the hardware qubit constraint while maintaining differentiability. First, to enforce the capacity limit, we employ a greedy capacity discretization (GCD) strategy. Unlike Gumbel-Softmax (Jang et al., 2017; Maddison et al., 2017), which fails to guarantee strict constraints, GCD iteratively assigns nodes based on descending probabilities to ensure $\mathbf{S} = \text{GCD}(\tilde{\mathbf{S}}) \in \{0, 1\}^{N \times k}$ respects the qubit limit (see Appendix A.2 for GCD details). Second, to enable back-propagation through the non-differentiable GCD, we apply the straight-through estimator (STE, Bengio, 2013):

$$\nabla_{\tilde{\mathbf{S}}} f \approx \nabla_{\mathbf{S}} f. \tag{10}$$

In the forward pass, we use the valid discrete $\mathbf{S}$ for evaluation; in the backward pass, we bypass the quantization error and propagate gradients directly to $\tilde{\mathbf{S}}$. Finally, we apply the mask $\mathbf{A}_{\text{sub}} = \mathbf{A} \odot (\mathbf{S}\mathbf{S}^\top)$ to isolate subgraph topologies for the subsequent parameter generation stage.

**Parameter Generation.** To model $P(\mathbf{P}|\mathbf{S}, G)$, we generate QAOA parameters conditioned on the discrete partition. Crucially, to prevent parameter optimization from destabilizing the learned partition structure, we apply a stop-gradient (sg) operator to the input topology. We first extract node features $\mathbf{H}_{\text{partition}} = \text{Encoder}_{\text{partition}}(\mathbf{X}, \text{sg}(\mathbf{A}_{\text{sub}}))$ and pool

them into subgraph embeddings $\mathbf{H}_{\text{sub}} = \text{GMP}(\mathbf{H}_{\text{partition}}) \in \mathbb{R}^{h \times k}$. Finally, to respect $2\pi$-periodicity, an MLP maps these embeddings to Cartesian coordinates, followed by a differentiable arctangent transformation that converts them to the variational parameters $\mathbf{P} \in [0, 2\pi]^{2p \times k}$.

# 5. Experiments

## 5.1. Experimental Setup

**Datasets.** We evaluate Neural QAOA² on a public benchmark library[1], which includes QUBO (datasets *B*, *BE*, *GKA*), Ising spin glass (datasets *L*, *BMZ*), and MaxCut (datasets *W*, *HR*) instances. All datasets contain multiple problem instances, with problem sizes $N$ exceeding the maximum qubit limit in our experiments (max_nodes = 10).

**Training Details.** We select 80% of instances from datasets *B*, *BE*, and *W* (problem sizes range from 51 to 501), which consist of QUBO and MaxCut instances only, to form the training graph set $\mathcal{D}_{\text{graph}}$. Ising spin glass instances are excluded from training. To construct the labeled dataset $\mathcal{D}_{\text{offline}} = \{(G_i, \mathbf{S}_i, \mathbf{P}_i, \rho_i)\}$ for the quantum evaluator, we augment each $G_i \in \mathcal{D}_{\text{graph}}$ with partitions $\mathbf{S}_i$ derived from heuristic baselines and parameters $\mathbf{P}_i$ sampled from a uniform distribution, obtaining the ground-truth performance ratio $\rho_i$ via QAOA² simulation. The quantum evaluator $f_\phi$ is first trained via supervised learning on $\mathcal{D}_{\text{offline}}$. Subsequently, the joint generator $g_\theta$ is trained on $\mathcal{D}_{\text{graph}}$ via gradient ascent, guided by the frozen $f_\phi$. Dataset details and hyperparameters are provided in Appendix B.

**Baselines.** We benchmark our method against the standard QAOA² framework equipped with four representative partitioning heuristics. We include the two primary heuristics from the original QAOA² study (Zhou et al., 2023): (a) **Random**, which assigns nodes to subgraphs stochastically, and (b) **Modularity** (Clauset et al., 2004), which maximizes graph modularity. Additionally, we evaluate (c) **Boundary** (Guerreschi, 2021) and (d) **KL** (Kernighan & Lin, 1970), which minimize the number of boundary nodes and the cut size between partitions, respectively. Following the protocol of the original QAOA² study (Zhou et al., 2023), all baselines employ random initialization for variational parameters, sampled from a uniform distribution $\mathcal{U}[0, 2\pi]$, with the quantum circuit depth $p$ set to 1. We further benchmark against advanced parameter initialization strategies including INTERP (Zhou et al., 2020), FOURIER (Zhou et al., 2020), TQA (Sack & Serbyn, 2021), and QIBPI (Katial et al., 2025), as well as deeper circuits ($p = 2, 3$). These additional results are provided in Appendix C.1.

**Metrics.** We evaluate performance using the average performance ratio ($\bar{\rho}$) and average rank. For each problem

---

[1]http://bqp.cs.uni-bonn.de/library/html/instances.html

*Table 1.* **Main Results on 50 Test Instances.** Cell layout: mean $\bar{\rho}$ (top), $\pm$ std (middle), and average rank with w/l counts (bottom) across different datasets (# denotes the instance count). **Bold** indicates the best performance (mean $\bar{\rho} \uparrow$, rank $\downarrow$). Neural QAOA² consistently outperforms heuristics, achieving the best average rank of **1.74** overall.

| Dataset (#) | QAOA² with Heuristic Partition | | | | Neural QAOA² |
|---|---|---|---|---|---|
| | Random | Modularity | Boundary | KL | GEN |
| *B* (8) | 0.8047 | 0.8351 | 0.8246 | 0.8092 | **0.8417** |
| | $\pm$ 0.0226 | $\pm$ 0.0304 | $\pm$ 0.0326 | $\pm$ 0.0177 | $\pm$ 0.0256 |
| | (4.75) 0/8 | (2.38) 2/6 | (2.63) 1/7 | (3.75) 0/8 | **(1.50)** 5/3 |
| *BE* (16) | 0.8626 | 0.8692 | 0.8722 | 0.8672 | **0.8824** |
| | $\pm$ 0.0344 | $\pm$ 0.0342 | $\pm$ 0.0359 | $\pm$ 0.0342 | $\pm$ 0.0304 |
| | (4.81) 0/16 | (3.13) 0/16 | (2.31) 1/15 | (3.69) 0/16 | **(1.06)** 15/1 |
| *W* (26) | 0.8962 | 0.9137 | 0.9114 | 0.8934 | **0.9153** |
| | $\pm$ 0.0543 | $\pm$ 0.0304 | $\pm$ 0.0335 | $\pm$ 0.0537 | $\pm$ 0.0280 |
| | (3.23) 2/24 | **(2.23)** 9/17 | (2.96) 6/20 | (4.27) 1/25 | (2.23) 8/18 |
| Overall (50) | 0.8708 | 0.8869 | 0.8850 | 0.8716 | **0.8930** |
| | $\pm$ 0.0552 | $\pm$ 0.0437 | $\pm$ 0.0465 | $\pm$ 0.0529 | $\pm$ 0.0390 |
| | (3.98) 2/48 | (2.54) 11/39 | (2.70) 8/42 | (4.00) 1/49 | **(1.74)** 28/22 |

*Table 2.* **Distribution Generalization on 93 Test Instances.** We evaluate on two out-of-distribution graph topologies: *GKA* (QUBO) and *L* (Ising spin glass). Cell layout: mean $\bar{\rho}$ (top), $\pm$ std (middle), and average rank with w/l counts (bottom). **Bold** indicates the best metric. Neural QAOA² achieves the best average rank of **1.46** overall.

| Dataset (#) | QAOA² with Heuristic Partition | | | | Neural QAOA² |
|---|---|---|---|---|---|
| | Random | Modularity | Boundary | KL | GEN |
| *GKA* (45) | 0.8478 | 0.8659 | 0.8601 | 0.8503 | **0.8762** |
| | $\pm$ 0.0441 | $\pm$ 0.0364 | $\pm$ 0.0414 | $\pm$ 0.0393 | $\pm$ 0.0359 |
| | (4.16) 2/43 | (2.40) 7/38 | (2.89) 4/41 | (4.04) 0/45 | **(1.51)** 32/13 |
| *L* (48) | 0.6984 | 0.7391 | **0.8205** | 0.7022 | 0.8160 |
| | $\pm$ 0.0305 | $\pm$ 0.0446 | $\pm$ 0.0440 | $\pm$ 0.0235 | $\pm$ 0.0285 |
| | (4.65) 0/48 | (3.06) 0/48 | (1.60) 20/28 | (4.27) 0/48 | **(1.42)** 28/20 |
| Overall (93) | 0.7707 | 0.8005 | 0.8397 | 0.7739 | **0.8451** |
| | $\pm$ 0.0837 | $\pm$ 0.0754 | $\pm$ 0.0471 | $\pm$ 0.0807 | $\pm$ 0.0441 |
| | (4.41) 2/91 | (2.74) 7/86 | (2.23) 24/69 | (4.16) 0/93 | **(1.46)** 60/33 |

instance, $\bar{\rho}$ is calculated as the average $\rho$ over 10 independent quantum simulation runs. Methods are then ranked on each instance based on their respective $\bar{\rho}$, and we report the mean rank averaged across all testing instances. Higher $\bar{\rho}$ and lower ranks indicate better performance.

**Code Availability.** We release the source code of our implementation[2] to facilitate reproducibility and future research.

### 5.2. Main Results

Table 1 compares Neural QAOA² with heuristic baselines on 50 held-out test instances, drawn from the remaining 20% of the *B*, *BE*, and *W* datasets, with distributions and problem scales consistent with the training set. In addition to the mean performance ratio $\bar{\rho}$, standard deviation (std), and average rank, we also report the win/loss (w/l) count, which measures the number of instances on which a method attains the best performance (rank 1). Overall, while the gains are not uniformly dominant across every single metric and dataset, Neural QAOA² demonstrates strong and relatively consistent performance, proving to be broadly competitive and often the best method. It achieves the best average rank of 1.74, winning on 28 out of 50 test instances.

**Performance on QUBO Problems.** Neural QAOA² exhibits particularly strong performance on QUBO instances. On the *BE* dataset, it achieves an average rank of 1.06, winning 15 out of 16 instances. In contrast, modularity performs less favorably (rank 3.13), as general QUBO problems often lack the explicit community structures that such heuristics exploit. By learning partitioning strategies guided by the predicted quantum optimization performance, our GEN effectively aligns partitioning decisions with the underlying Hamiltonian landscape.

**Performance on MaxCut.** On the *W* dataset (MaxCut),

---

[2]https://github.com/0SliverBullet/Neural-QAOA-Squared

which naturally exhibits graph clustering structure, modularity is particularly strong. Neural QAOA² is tied with this strongest heuristic baseline (both achieving an average rank of 2.23) without relying on handcrafted graph-theoretic rules, indicating that the GEN successfully captures relevant topological properties through data-driven learning.

**Stability.** Beyond average performance, Neural QAOA² demonstrates improved stability across test instances. As reflected by lower standard deviations, the learned generator adapts to diverse graph structures, mitigating the high variance often observed in heuristics that are sensitive to specific topologies.

Notably, our GEN is trained exclusively at $p = 1$. For deeper circuits ($p > 1$), we employ the standard parameter expansion strategy from prior work (Zhou et al., 2020) rather than retraining a high-depth model from scratch. As demonstrated in Appendix C.1, our method maintains competitive performance at $p = 2$ and $p = 3$, even outperforming QAOA² equipped with stronger initialization baselines (TQA, INTERP, FOURIER, and QIBPI) overall.

### 5.3. Distribution Generalization Analysis

To assess zero-shot generalization, we evaluate Neural QAOA² on 93 instances from unseen distributions. These instances share comparable problem sizes ($N \in [21, 501]$) with the training set but possess distinct topologies. The test suite comprises 45 *GKA* (QUBO) and 48 *L* (Ising spin glass) instances. Results in Table 2 indicate effective generalization: Neural QAOA² achieves the best overall average rank of 1.46 and the highest win rate (60/93).

**Performance on QUBO Problems.** Our method performs best on general QUBO instances, achieving the highest mean $\bar{\rho}$ (0.8762) and winning 32 out of 45 instances. It outperforms modularity (rank 2.40) and boundary (rank 2.89), highlighting the effectiveness of the generated partition.

**Performance on Ising Spin Glass.** This class of instances

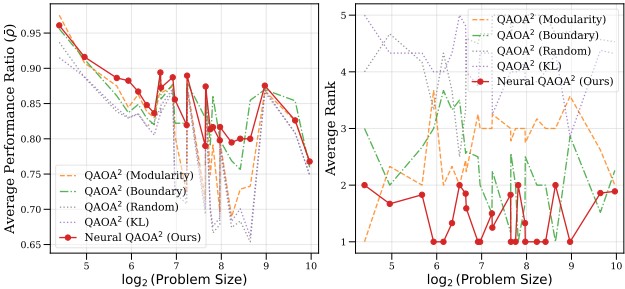

*Figure 3.* **Problem Size Generalization.** We evaluate Neural QAOA² on test instances ranging from $N = 21$ to $N = 1000$ (plotted on a $\log_2$ scale). The left panel shows the mean $\bar{\rho}$, and the right panel shows the average rank. Our method (red line) maintains superior ranking stability across the spectrum, effectively generalizing to instances with scales unseen during training.

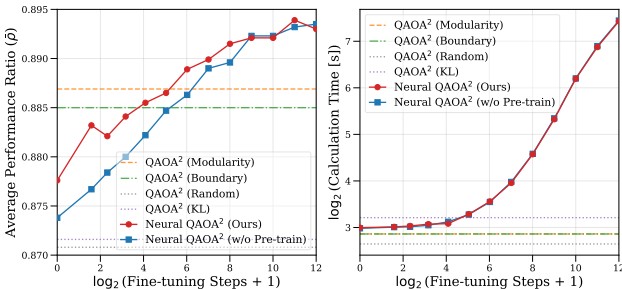

*Figure 4.* **Dynamics of Test-Time Adaptation.** Performance (left) and wall-clock time (right) vs. fine-tuning steps (plotted on a $\log_2$ scale). The red curve shows monotonic improvement, outperforming the strongest heuristic (modularity) after 64 steps. Comparison with the non-pretrained variant (blue) confirms that pre-training provides a critical initialization for rapid convergence.

is not seen during training. The boundary heuristic performs well here (mean $\bar{\rho}$ 0.8205), as minimizing boundary vertices aligns naturally with the lattice topology of spin glass graphs. In contrast, modularity, the runner-up on *GKA*, generalizes less effectively to these grid structures (mean $\bar{\rho}$ 0.7391). Neural QAOA² exhibits greater robustness: despite a slightly lower mean $\bar{\rho}$ (0.8160) than boundary, GEN achieves the best average rank (1.42) and secures the most wins (28/48), surpassing boundary (20 wins). We emphasize average rank and win count in our evaluation because they better capture consistency and mitigate the influence of performance variations on specific instances. This indicates that while heuristics such as boundary or modularity exhibit substantial performance variation depending on the graph topology, GEN consistently identifies high-quality partitions across diverse distributions.

### 5.4. Size Generalization Analysis

Size generalization requires a neural solver to handle instances across a broad spectrum of problem sizes unseen during training. To validate this, we extend our evaluation to a suite of 183 instances with problem sizes $N$ ranging from 21 to 1000. Supplementing the datasets in Sections 5.2 and 5.3, we incorporated two large-scale datasets: 30 Max-Cut instances from *HR* ($N = 800, 1000$) and 10 Ising spin glass instances from *BMZ* ($N = 1000$). This setup allows us to assess performance on scales seen during training ($51 \leq N \leq 501$) as well as unseen scales, including small-scale interpolation ($N < 51$) and large-scale extrapolation ($N > 501$). Figure 3 illustrates the performance trends.

**Performance on Seen Scales.** In the training range ($51 \leq N \leq 501$), Neural QAOA² establishes clear dominance. The mean $\bar{\rho}$ curve (Left, red line) consistently surpasses most baselines, and our method achieves the best average rank (rank 1) in 9 out of 21 evaluated size points.

**Performance on Unseen Scales.** Crucially, the model

demonstrates strong generalization capabilities on unseen scales. For interpolation ($N \in \{21, 31\}$), Neural QAOA² maintains a stable rank. For extrapolation ($N \in \{800, 1000\}$), where heuristics often struggle due to the expanding search space, our method maintains superior ranking stability. While absolute performance gaps naturally narrow at extreme scales, the consistently low rank indicates that GEN retains effective partitioning capability on graphs larger than those seen during training.

We also evaluate generalization across varying hardware qubit constraints (max_nodes) in Appendix C.2.

### 5.5. Ablation Studies

To verify the effectiveness of our Neural QAOA², we analyze the dynamics of test-time adaptation (TTA) and quantify the contribution of individual components.

**Effectiveness of Test-Time Adaptation.** Figure 4 (Left) illustrates the evolution of the mean $\bar{\rho}$ as fine-tuning steps increase from 0 to 4096. The monotonic upward trend validates the efficacy of the quantum evaluator. Although the evaluator remains fixed during inference, its backpropagated gradients provide informative guidance, effectively directing the joint generator to refine the partition **S** and parameters **P** for specific instances. Comparing the pre-trained model (red curve) against the non-pretrained variant (blue curve) reveals that pre-training the joint generator confers a significantly superior initialization (step 0), enabling Neural QAOA² to reach high-quality configurations with fewer update steps.

**Computational Efficiency.** A critical consideration for neural solvers is the trade-off between performance and inference latency. Figure 4 (Right) plots wall-clock time against fine-tuning steps. Note that these wall-clock comparisons include the quantum circuit simulation cost uniformly across all methods. While TTA incurs additional computational

*Table 3.* **Ablation Study of Neural QAOA² Components.** $\Delta\%$ denotes the relative performance drop compared to Neural QAOA². Statistical significance is measured via a one-sided paired t-test. Differences are considered statistically significant at $p < 0.01$.

| Variant (Fine-tuning Steps = 64) | Mean $\bar{\rho}$ | $\Delta\%$ | $p$-value |
|---|---|---|---|
| Neural QAOA² | 0.8892 | – | – |
| *Ablation on Quantum Evaluator Inputs* | | | |
| w/o Global Topology View | 0.8819 | -0.83% | $6.93 \times 10^{-4}$ |
| w/o Partition-Induced Subgraph View | 0.8797 | -1.07% | $7.44 \times 10^{-7}$ |
| w/o Quantum Parameter View | 0.8821 | -0.80% | $2.40 \times 10^{-4}$ |
| *Ablation on Joint Generator Mechanisms* | | | |
| w/o Orthogonal Complement Head | 0.8817 | -0.85% | $7.56 \times 10^{-5}$ |
| w/o Stop-Gradient | 0.8820 | -0.81% | $1.22 \times 10^{-4}$ |

cost, the overhead is acceptable compared to the substantial performance gains. In our practical inference protocol, we separate ceiling analysis from the recommended deployment budget: while adapting for 1,000 steps serves to estimate the empirical performance ceiling, 64 steps is defined as the default practical budget unless otherwise noted. Notably, at this default budget of 64 fine-tuning steps, Neural QAOA² achieves a mean $\bar{\rho}$ of 0.8892, surpassing the best heuristic (modularity, 0.8869) and all other baselines. Crucially, the total calculation time at this point is 11.78s, which remains within the same order of magnitude as modularity (7.27s), boundary (7.26s), and KL (9.26s).

**Component Contribution Analysis.** We conduct a comprehensive ablation study to isolate the contribution of each module. Table 3 summarizes the performance degradation ($\Delta\%$) of each variant relative to the full Neural QAOA². All observed performance drops are statistically significant ($p < 0.01$), suggesting that each component contributes.

First, regarding evaluator inputs, the partition-induced subgraph view proves most critical ($\Delta = -1.07\%$), confirming that perceiving subgraph topology is paramount for accurate performance prediction. The global topology and quantum parameter views contribute comparably ($\approx -0.8\%$), ensuring sufficient information encoding. Second, regarding generator mechanisms, replacing the OCH with unconstrained learnable cluster centers (i.e., removing the orthogonality constraints) degrades performance by $0.85\%$. This highlights that the dynamic orthogonalization in OCH is vital for enforcing inter-cluster separability. Finally, removing the stop-gradient operator causes a $0.81\%$ drop, verifying that decoupling the gradients of partition and parameter generation is essential to prevent optimization interference.

### 5.6. Study on the Effect of Noise

Quantum devices are subject to various types of noise, such as bit-phase flips, depolarization, amplitude damping, and phase damping. To evaluate the robustness of our method against realistic NISQ noise, we conduct experiments under noisy conditions using the 16 held-out *BE* instances

*Table 4.* **Performance Comparison under Different Noise Settings.** We examine shot noise (using 1,000 shots) and bit-phase flip errors with an error probability $p$ varying from 0 to 0.05.

| Noise Setting | Mean $\bar{\rho}$ |
|---|---|
| Noiseless | 0.8798 |
| Shot noise | 0.8733 |
| Bit-phase flip ($p = 0.01$) | 0.8780 |
| Bit-phase flip ($p = 0.05$) | 0.8763 |

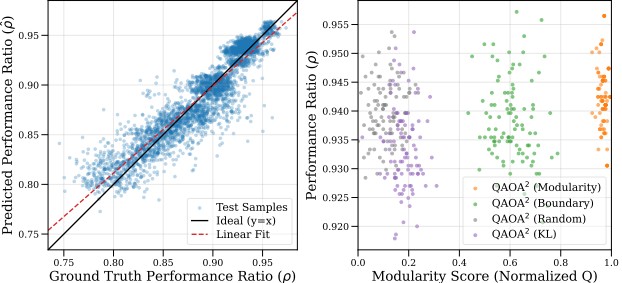

*Figure 5.* **Visual Analysis of Metric Misalignment.** Left: quantum evaluator predictions vs. ground truth across 3313 unseen test data points ($r = 0.9258$), showing high fidelity. Right: modularity vs. ground truth on the test instance *g05_100.1* ($r = 0.2859$), exposing a severe metric misalignment that lacks correlation with quantum performance.

from Section 5.2. Specifically, we introduce shot noise (using 1,000 shots) during the parameter optimization stage where it affects the gradient estimation. Additionally, following (Ye et al., 2023), we add bit-phase flips as readout noise at the end of the circuit with an error probability $p$ ranging from 0 to 0.05. The two noise models are evaluated separately. The experimental results are summarized in Table 4. As shown, the average performance ratio $\bar{\rho}$ experiences only slight degradation even at a 5% error rate, demonstrating that our method is not overly fragile to moderate noise.

### 5.7. A Deeper Look Into Why It Works

We empirically validate the two motivations outlined in Section 1: resolving heuristic metric misalignment and mitigating cold-start issues caused by topology-blind initialization.

**Resolving Metric Misalignment.** To verify the fidelity of our quantum evaluator against traditional heuristics, we analyze the correlation between predicted $\hat{\rho}$ and ground-truth performance ratio $\rho$. Specifically, we use the quantum evaluator to predict the performance ratio $\hat{\rho}$ for 3313 test configurations $(G_i, \mathbf{S}_i, \mathbf{P}_i)$, where each graph $G_i$ is drawn from the 50 held-out instances in *B*, *BE*, and *W* datasets. As shown in Figure 5 (Left), the predictions from our evaluator exhibit a strong linear correlation with the ground truth (Pearson's $r = 0.9258$), with data points closely adhering

*Table 5.* **Impact of Joint Parameter Initialization.** Comparison against a variant using random parameter initialization. Neural QAOA² (20 steps) outperforms random initialization even when the latter is given double the optimization budget (40 steps), confirming the effectiveness of our topology-aware warm start. Statistical significance is measured via a one-sided paired t-test. Differences are considered statistically significant at $p < 0.01$.

| Variant (Fine-tuning Steps = 64) | Mean $\bar{\rho}$ | $\Delta\%$ | Time [s] | $p$-value |
|---|---|---|---|---|
| *QAOA Optimization Steps = 20* | | | | |
| Neural QAOA² | 0.8892 | – | 11.78 | – |
| Neural QAOA² (Random **P**) | 0.8833 | -0.67% | 11.75 | $1.89 \times 10^{-5}$ |
| *QAOA Optimization Steps = 40* | | | | |
| Neural QAOA² (Random **P**) | 0.8863 | -0.34% | 17.49 | $8.22 \times 10^{-3}$ |

to the $y = x$ identity line. This confirms that the evaluator acts as a high-fidelity differentiable proxy, accurately guiding the generator toward goal-aligned partitions. In contrast, Figure 5 (Right) reveals that modularity, a widely used graph partitioning metric, shows no significant correlation with quantum solution quality (Pearson's $r = 0.2859$) on the representative test instance *g05_100.1*. The scatter plot resembles a random distribution, empirically proving that maximizing modularity does not guarantee better quantum optimization performance. This metric misalignment fundamentally explains why heuristic baselines often fail to identify high-quality partitions for QAOA². We further assess evaluator calibration on generator-produced configurations during TTA and observe consistent alignment with ground truth performance (Appendix C.4).

**Mitigating Cold Start.** To quantify the benefit of our topology-aware parameter initialization, we isolate the impact of the generated parameters **P**. We compare Neural QAOA² against a variant that uses the same learned partition **S** but reverts to random parameter initialization ($\gamma, \beta \sim \mathcal{U}[0, 2\pi]$). As reported in Table 5, under the same optimization budget (20 steps), random initialization incurs a statistically significant performance drop ($\Delta = -0.67\%$). Even when doubling the optimization budget to 40 steps, the random initialization variant still fails to match our method (0.8863 vs. 0.8892), while incurring an additional $\approx 48\%$ computational overhead (17.49s vs. 11.78s). This result indicates that our generated parameters are not arbitrary; rather, the joint generator learns an intrinsic mapping from subgraph topology to favorable basins of attraction in the parameter landscape, conditioned on the partition. By providing a warm start, it effectively avoids the poor local minima inherent to random initialization.

## 6. Conclusion

We present Neural QAOA², an end-to-end differentiable framework for scalable quantum combinatorial optimization. By integrating a generative evaluative network (GEN), our approach aligns partitioning decisions with quantum ob-

jectives and alleviates optimization cold starts via topology-aware initialization. Through specialized differentiable mechanisms, GEN enables gradient-based learning over discrete graph structures under strict qubit capacity constraints. Extensive experiments across diverse benchmarks show consistent improvements over heuristic baselines and strong generalization to unseen distributions and scales.

## 7. Limitations

Despite these advancements, our current evaluation is limited to noiseless simulation. Future work will incorporate hardware-specific noise models directly into the quantum evaluator. Additionally, while our framework effectively optimizes QAOA²-compatible settings, its mechanism inherently relies on the unconstrained binary structure and $\mathbb{Z}_2$ symmetry, which limits its direct application to strongly constrained problems like TSP or MIS. Extending gradient-driven divide-and-conquer paradigms to handle such hard constraints remains a promising future direction. Furthermore, exploring meta-learning approaches (e.g., MAML) to unify our two-stage distribution-level learning presents an exciting avenue (see Appendix F). More broadly, our approach depends on an offline dataset, although the acquisition cost is relatively low (see Appendix B.2); reinforcement learning may offer an alternative training paradigm.

## Acknowledgements

This work was supported by National Key Research and Development Program of China under Grant 2022YFA1004102, and in part by National Natural Science Foundation of China under Grant 62502192, and in part by National Natural Science Foundation of China under Grant 62272210.

## Impact Statement

This paper presents work whose goal is to advance the field of Machine Learning. Specifically, our work aims to empower quantum computing via classical deep learning. By introducing a generative evaluative network to jointly generate graph partitioning and initialize quantum circuit parameters, we significantly enhance the scalability and performance of quantum algorithms for solving large-scale combinatorial optimization problems. We have open-sourced our method, which will contribute to the community of quantum machine learning and optimization research. There are many potential societal consequences of our work, none which we feel must be specifically highlighted here.

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

# A. Methodological Details

## A.1. Orthogonal Complement Head (OCH) Implementation

In this section, we provide the implementation details of the orthogonal complement head (OCH). As introduced in Section 4.3, OCH generates the cluster center matrix $\mathbf{C} \in \mathbb{R}^{k \times h}$ deterministically to satisfy two geometric constraints: (1) Cluster centers must be orthogonal to the global mean ($c_i g = 0$), ensuring clustering is based on structural deviations rather than commonality; (2) Cluster centers must be mutually orthogonal ($c_i c_j^\top = 0, \forall i \neq j$), where $c_i$ is the $i$-th row of $\mathbf{C}$.

Let $\mathbf{H}_{\text{topology}} \in \mathbb{R}^{N \times h}$ denote the node embeddings generated by the topology encoder. The construction of the soft partition $\tilde{\mathbf{S}}$ follows a four-step process:

**Step 1: Global Context Normalization.** First, we compute the global context vector $g \in \mathbb{R}^h$ to capture the dominant direction of the graph embeddings. To ensure numerical stability and directional consistency, we utilize the normalized mean of normalized embeddings:

$$g = \frac{\tilde{g}}{\|\tilde{g}\|_2}, \quad \text{where } \tilde{g} = \frac{1}{N} \sum_{i=1}^{N} \frac{h_i^\top}{\|h_i\|_2}. \tag{11}$$

Here $h_i$ is the $i$-th row of $\mathbf{H}_{\text{topology}}$. The resulting $g$ lies on the unit hypersphere $\mathbb{S}^{h-1}$.

**Step 2: Stochastic Anchor Sampling.** To generate diverse basis vectors orthogonal to $g$, we maintain a frozen pool of stochastic anchors $\mathbf{P}_{\text{pool}} \in \mathbb{R}^{k_{\max} \times h}$, sampled from a standard normal distribution $\mathcal{N}(0, 1)$ at initialization. For a target subgraph number $k$, we slice the first $k$ vectors from the pool to form the auxiliary anchor matrix $\mathbf{A}_k \in \mathbb{R}^{k \times h}$. Using a fixed ordered slice ensures that the subspace for a smaller $k$ is a strict subset of the subspace for a larger $k$, stabilizing training dynamics.

**Step 3: Orthogonal Basis Generation via QR Decomposition.** We construct a raw basis matrix $\mathbf{M} \in \mathbb{R}^{h \times (k+1)}$ by augmenting the global context vector $g$ with the transpose of the anchor matrix $\mathbf{A}_k$:

$$\mathbf{M} = \left[ g \mid \mathbf{A}_k^\top \right] = \left[ g \mid a_1^\top, \ldots, a_k^\top \right], \tag{12}$$

where $g$ serves as the first column, and $a_i$ is the $i$-th row of $\mathbf{A}_k$. We then perform a reduced QR decomposition on $\mathbf{M}$:

$$\mathbf{M} = \mathbf{QR}, \quad \text{where } \mathbf{Q} \in \mathbb{R}^{h \times (k+1)} \text{ and } \mathbf{Q}^\top \mathbf{Q} = \mathbf{I}. \tag{13}$$

Due to the sequential nature of the Gram-Schmidt process inherent in QR decomposition (Golub & Van Loan, 2013), the first column of $\mathbf{Q}$, denoted as $q_0$, aligns with $g$ (i.e., $q_0 = g$). The subsequent columns $\{q_1, \ldots, q_k\}$ span a subspace that is strictly orthogonal to $g$ and mutually orthonormal.

**Step 4: Cluster Center Extraction and Soft Partition.** We discard the first column $q_0$ (which corresponds to the global mean) and collect the remaining orthogonal vectors to form the cluster center matrix $\mathbf{C}$:

$$\mathbf{C} = [q_1, \ldots, q_k]^\top \in \mathbb{R}^{k \times h}. \tag{14}$$

By construction, the rows of $\mathbf{C}$ satisfy:

- Orthogonality to global context: $c_i g = q_i^\top q_0 = 0$ (since columns of $\mathbf{Q}$ are orthogonal).

- Mutual orthogonality: $c_i c_j^\top = q_i^\top q_j = \delta_{ij}$ (implying $\mathbf{CC}^\top = \mathbf{I}$).

Finally, the soft partition matrix $\tilde{\mathbf{S}} \in [0, 1]^{N \times k}$ is computed via softmax similarity between node embeddings and generated cluster centers:

$$\tilde{\mathbf{S}}_{ij} = \frac{\exp(h_i c_j^\top / \tau)}{\sum_{m=1}^{k} \exp(h_i c_m^\top / \tau)}, \tag{15}$$

where $\tau$ is a temperature hyperparameter that controls the sharpness of the assignment distribution. This process ensures that the partitioning is driven purely by the structural deviations orthogonal to the common global mean.

## A.2. Greedy Capacity Discretization (GCD) Algorithm

To strictly enforce the hardware constraint where each subgraph cannot exceed a maximum number of qubits (max_nodes), we propose the greedy capacity discretization (GCD) algorithm. Unlike global sorting strategies that are computationally expensive and hard to parallelize, GCD employs a row-wise decoupled sorting followed by a round-based greedy negotiation strategy.

---

**Algorithm 1** Greedy Capacity Discretization (GCD)

---

1: **Input:** Soft partition matrix $\tilde{\mathbf{S}} \in [0,1]^{N \times k}$, Qubit capacity limit $C_{\max} = \text{max\_nodes}$
2: **Output:** Discrete partition matrix $\mathbf{S} \in \{0,1\}^{N \times k}$
3: Initialize $\mathbf{S} \leftarrow \{0\}^{N \times k}$, unassigned nodes set $\mathcal{V}_{\text{unassigned}} \leftarrow \{1, \ldots, N\}$, partition loads $\mathbf{L} \leftarrow [0, \ldots, 0] \in \mathbb{R}^k$
4: Parallel Step: For each node $i$, sort row $\tilde{\mathbf{S}}_{i,:}$ to get ranks $\mathcal{R}_i = [r_{i,1}, \ldots, r_{i,k}]$
5: **for** priority round $l = 1$ **to** $k$ **do**
6:     Clear candidate lists $\mathcal{U}_j \leftarrow \emptyset$ for all $j \in \{1, \ldots, k\}$
7:     **for** each node $i \in \mathcal{V}_{\text{unassigned}}$ **do**
8:         Target partition $j \leftarrow r_{i,l}$
9:         Add $i$ to $\mathcal{U}_j$
10:     **end for**
11:     **for** each partition $j \in \{1, \ldots, k\}$ **do**
12:         Capacity remaining $C_{\text{remain}}^{(j)} \leftarrow C_{\max} - \mathbf{L}_j$
13:         **if** $|\mathcal{U}_j| \leq C_{\text{remain}}^{(j)}$ **then**
14:             Accept set $\mathcal{A} \leftarrow \mathcal{U}_j$
15:         **else**
16:             Sort nodes in $\mathcal{U}_j$ by score $\tilde{\mathbf{S}}_{i,j}$ descending
17:             Accept set $\mathcal{A} \leftarrow \text{Top-}C_{\text{remain}}^{(j)}(\mathcal{U}_j)$
18:         **end if**
19:         **for** each node $i \in \mathcal{A}$ **do**
20:             $\mathbf{S}_{i,j} \leftarrow 1$
21:             $\mathbf{L}_j \leftarrow \mathbf{L}_j + 1$
22:             Remove $i$ from $\mathcal{V}_{\text{unassigned}}$
23:         **end for**
24:     **end for**
25:     **Break** if $\mathcal{V}_{\text{unassigned}}$ is empty
26: **end for**

---

Let $\tilde{\mathbf{S}} \in [0,1]^{N \times k}$ be the soft partition probabilities from the OCH. The process involves two phases:

**Phase 1: Row-wise Decoupled Sorting.** Instead of sorting all $N \times k$ elements globally, we decouple the problem into $N$ independent sorting tasks. For each node $i$, we sort its partition probabilities in descending order to obtain a preference list:

$$\mathcal{R}_i = [r_{i,1}, r_{i,2}, \ldots, r_{i,k}], \quad \text{s.t. } \tilde{\mathbf{S}}_{i,r_{i,1}} \geq \tilde{\mathbf{S}}_{i,r_{i,2}} \cdots \geq \tilde{\mathbf{S}}_{i,r_{i,k}}. \tag{16}$$

This step identifies the 1st-choice, 2nd-choice, ..., $k$-th choice partition for every node.

**Phase 2: Round-based Greedy Negotiation.** We simulate a multi-round admission process iterating from priority level $l = 1$ to $k$:

1. Candidate selection: In round $l$, any unassigned node $i$ applies to its $l$-th preferred partition $j = r_{i,l}$. We collect all applicants for partition $j$ into a candidate set $\mathcal{U}_j$.

2. Conflict resolution: We check the remaining capacity of partition $j$, denoted as $C_{\text{remain}}^{(j)}$.

   - If $|\mathcal{U}_j| \leq C_{\text{remain}}^{(j)}$, all candidates are accepted.

- If $|\mathcal{U}_j| > C_{\text{remain}}^{(j)}$, we sort candidates by their confidence scores (probability values $\tilde{\mathbf{S}}_{i,j}$) and accept the top-$C_{\text{remain}}^{(j)}$ nodes.

3. State update: Accepted nodes are locked into partition $j$. Rejected nodes remain unassigned and automatically proceed to round $l+1$ to attempt their next preferred partition.

**Complexity Analysis and Parallel Efficiency.**    We acknowledge that the theoretical worst-case complexity of GCD is dominated by the conflict resolution in Phase 2. In a pathological scenario where node preferences are highly centralized (e.g., all $N$ nodes contend for the same partition in every round), the sorting in Line 16 requires $\mathcal{O}(N \log N)$ operations. Iterated over $k$ rounds, the strict worst-case bound is $\mathcal{O}(kN \log N)$, which is asymptotically equivalent to global sorting strategies.

However, the practical computational advantage of GCD stems from its decoupling strategy and hardware efficiency, rather than a theoretical reduction in the worst-case bound:

- Parallelism in Phase 1: The sorting in Phase 1 has a complexity of $\mathcal{O}(Nk \log k)$ and is strictly row-independent. This allows us to map the task to $N$ parallel GPU threads via CUDA, thereby substantially reducing wall-clock time compared to global sorting which is harder to parallelize efficiently.

- Average-case performance: In practice, the soft partition probabilities $\tilde{\mathbf{S}}$ generated by the OCH are trained to be discriminative, meaning candidates are typically distributed across different partitions. This effectively avoids the worst-case collision scenario in Phase 2, maintaining high inference efficiency.

The complete procedure is summarized in Algorithm 1.

**Differentiability via Straight-Through Estimator (STE).**    Since the GCD operation involves non-differentiable sorting and thresholding, standard backpropagation cannot be directly applied. To enable end-to-end training, we employ the straight-through estimator (STE, Bengio, 2013). During the forward pass, we use the discrete partition matrix $\mathbf{S} = \text{GCD}(\tilde{\mathbf{S}})$. During the backward pass, we bypass the discretization operations and approximate the gradients as identity:

$$\nabla_{\tilde{\mathbf{S}}} f \approx \nabla_{\mathbf{S}} f. \tag{17}$$

While the STE has historically been regarded as a heuristic, recent theoretical work provides a principled interpretation of its effectiveness for training models with discrete components. In particular, Liu et al. (2023a) show that the original STE can be viewed as a special case of the forward Euler discretization of an ordinary differential equation associated with the smoothing of discrete functions. Under this perspective, the STE corresponds to a first-order approximation of the underlying gradient flow, offering a theoretical explanation for why it can yield meaningful descent directions despite discontinuities in the forward mapping.

Beyond this general interpretation, the effectiveness of the STE in our GCD module further relies on a problem-specific structural property. In our construction, the discretization step exhibits a monotonic relationship between the continuous scores $\tilde{\mathbf{S}}$ and the resulting discrete assignments $\mathbf{S}$: increasing $\tilde{\mathbf{S}}_{ij}$ improves the rank of node $i$ within partition $j$, thereby increasing the likelihood of $\mathbf{S}_{ij}$ being set to 1. This positive correlation implies that, although the gradient estimator is biased, the sign of the STE gradient remains aligned with improvements in the discrete partitioning objective, effectively guiding the continuous parameters $\theta$ toward favorable discrete configurations.

**Comparison with Differentiable Relaxations.**    We deliberately employ GCD with STE over alternative differentiable relaxation methods such as the Gumbel-Sinkhorn networks (Mena et al., 2018) or differentiable projection layers (e.g., OptNet (Amos & Kolter, 2017), CvxpyLayers (Agrawal et al., 2019)) due to the strict necessity of hard constraints imposed by quantum hardware resources.

Relaxation methods operate over continuous assignment spaces, producing soft assignment matrices (e.g., doubly stochastic matrices in Sinkhorn) where entries are continuous probabilities $p_{ij} \in [0, 1]$. While differentiable, they suffer from three critical limitations:

- Violation of hard constraints: Soft assignments require post-hoc rounding (e.g., argmax) to obtain discrete decisions. This rounding process often destroys constraint guarantees. A soft assignment satisfying $\sum_i p_{ij} \leq C_{\max}$ does not guarantee that the rounded discrete subgraph size $\sum_i \mathbb{I}(j = \text{argmax}_k p_{ik})$ respects the hardware qubit limit $C_{\max}$.

- Incompatibility with inequalities: Methods like Sinkhorn are naturally formulated for equality constraints (fixed marginals), making them ill-suited for the inequality constraints ($\leq C_{\max}$) intrinsic to partitioning.

- Computational overhead: Rigorous rounding strategies for constrained problems (e.g., via the Hungarian algorithm) typically incur cubic complexity $\mathcal{O}(N^3)$ (Kuhn, 1955), limiting scalability to large graphs.

In contrast, our GCD algorithm incorporates the capacity constraint $C_{\max}$ (e.g., max_nodes=10) directly into the forward discretization pass with efficient parallel logic ($\mathcal{O}(Nk \log k)$). By enforcing these constraints strictly during the forward pass, we ensure that the quantum evaluator always receives feasible subgraphs that can be physically mapped to the quantum device. This eliminates the validity gap often observed in relaxation-based methods where the optimized soft assignment projects to an infeasible discrete assignment.

### A.3. Network Architecture Specifications

In this section, we detail the specific architectures of the neural components utilized in Neural QAOA². The GEN framework employs two primary backbone encoders: the GAT encoder (based on GATv2 (Brody et al., 2022)) and the GCN encoder (based on standard GCN (Kipf & Welling, 2017)), which are reused across different modules.

**Input Feature Initialization.** To capture the intrinsic structural properties of the input graph while maintaining permutation equivariance, we explicitly construct the initial node features $\mathbf{X} \in \mathbb{R}^{N \times d}$. Specifically, for each node $i$, the input feature vector $\boldsymbol{x}_i$ consists of five standardized structural descriptors (i.e., $d = 5$):

$$\boldsymbol{x}_i = [d_i, \, s_i, \, C_i, \, \mathrm{PR}_i, \, \mathrm{BC}_i], \tag{18}$$

where $d_i$ is node degree, $s_i$ is weighted degree, $C_i$ is clustering coefficient, $\mathrm{PR}_i$ is PageRank score, and $\mathrm{BC}_i$ is betweenness centrality. These features collectively encode both local connectivity and global topological influence, and are normalized to zero mean and unit variance to ensure numerical stability for the subsequent GNN encoders. The mathematical definitions are provided below, where $\mathbf{A}_{ij}$ directly denotes the weighted adjacency matrix (i.e., $\mathbf{A}_{ij}$ carries the edge weight value, and $\mathbf{A}_{ij} = 0$ implies no edge):

- Node degree & weighted degree: The node degree (unweighted) is defined as the number of neighbors, $d_i = \sum_j \mathbb{I}(\mathbf{A}_{ij} \neq 0)$, where $\mathbb{I}(\cdot)$ is the indicator function. The weighted degree (strength) is the sum of incident edge weights, defined as $s_i = \sum_j \mathbf{A}_{ij}$.

- Clustering coefficient: We utilize the weighted variant adapted for signed graphs by considering the absolute weights $|\mathbf{A}_{ij}|$:

$$C_i = \frac{1}{d_i(d_i - 1)} \sum_{j,k} (\hat{\mathbf{A}}_{ij} \hat{\mathbf{A}}_{jk} \hat{\mathbf{A}}_{ki})^{1/3}, \tag{19}$$

where $\hat{\mathbf{A}}_{ij} = |\mathbf{A}_{ij}| / \max(|\mathbf{A}_{ij}|)$ represents the normalized absolute edge weights (Onnela et al., 2005; Saramäki et al., 2007).

- PageRank score: To quantify the global influence, we compute the PageRank score $\mathrm{PR}_i$ using absolute weights to ensure convergence on signed graphs (Page et al., 1999):

$$\mathrm{PR}_i = \alpha \sum_{j \in \mathcal{N}(i)} \frac{|\mathbf{A}_{ji}|}{D_j^{\mathrm{out}}} \mathrm{PR}_j + (1 - \alpha) \frac{1}{N}, \tag{20}$$

where $\alpha$ is the damping factor (set to 0.85) and $D_j^{\mathrm{out}} = \sum_k |\mathbf{A}_{jk}|$ is the out-degree sum of absolute weights.

- Betweenness centrality: This metric quantifies the influence of a node as a bridge along shortest paths (Freeman, 1977). It is defined as:

$$\mathrm{BC}_i = \sum_{s \neq i \neq t} \frac{\sigma_{st}(i)}{\sigma_{st}}, \tag{21}$$

where $\sigma_{st}$ is the total number of shortest paths from node $s$ to $t$ (computed using $|\mathbf{A}_{ij}|$ as distance costs), and $\sigma_{st}(i)$ is the number of those paths passing through node $i$.

Note on feature interpretation: We explicitly interpret the edge weight $|\mathbf{A}_{ij}|$ as a resistance or transport cost. Under this formulation, shortest paths favor edges with lower weights (weaker interactions). Consequently, this metric identifies nodes that serve as critical bridges across "weakly-coupled" regions, providing the GNN with structural bottleneck information that complements the direct connectivity signals encoded in the adjacency matrix.

*Table 6.* Architecture specifications for the Quantum Evaluator.

| Component | Backbone | Input Specification |
|---|---|---|
| Topology Encoder | GAT encoder (Hidden Dim: 64, Layers: 3) | Node features $\mathbf{X} \in \mathbb{R}^{N\times 5}$, Adjacency $\mathbf{A}$ |
| Partition Encoder | GCN encoder (Hidden Dim: 64, Layers: 3) | Node features $\mathbf{X}$, Masked Adjacency $\mathbf{A}_{\text{sub}}$ |
| Parameter Encoder | GAT encoder (Hidden Dim: 64, Layers: 3) | Sin-Cos Embeddings $\tilde{\mathbf{X}}_{\text{param}} \in [-1, 1]^{N\times 4p}$, where $p = 1$ |
| GMP | Global Mean Pooling $\rightarrow$ Concatenation (Dim: 192) | |
| MLP | MLP: Linear$(192, 256) \rightarrow$ ReLU $\rightarrow$ Linear$(256, 1) \rightarrow$ Sigmoid | |

*Table 7.* Architecture specifications for the Joint Generator.

| Component | Backbone | Input Specification |
|---|---|---|
| *Partition Generator* | | |
|   Topology Encoder | GAT encoder (Hidden Dim: 128, Layers: 2) | Node features $\mathbf{X} \in \mathbb{R}^{N\times 5}$, Adjacency $\mathbf{A}$ |
|   OCH | OCH ($k_{\max}$ : 127, Hidden Dim: 128, $\tau$: 0.05) | Node Embeddings $\mathbf{H}_{\text{topology}} \in \mathbb{R}^{N\times 128}$ |
| *Parameter Generator* | | |
|   Partition Encoder | GCN encoder (Hidden Dim: 128, Layers: 2) | Node features $\mathbf{X}$, Masked Adjacency $\mathbf{A}_{\text{sub}}$ |
|   GMP | Global Mean Pooling $\rightarrow \mathbf{H}_{\text{sub}} \in \mathbb{R}^{128\times k}$ | |
|   MLP | Linear$(128, 128) \rightarrow$ReLU$\rightarrow$ Linear$(128, 4p) \rightarrow \arctan 2 \rightarrow \mathbf{P} \in [0, 2\pi)^{2p\times k}$, where $p = 1$ | |

**Shared Backbone Architectures.** The structural specifications for the shared encoders are as follows:

- GAT encoder: This module first projects initial node features into a latent space via a feature embedding block: Linear$(d, h) \rightarrow$ ReLU $\rightarrow$ Linear$(h, h) \rightarrow$ ReLU. This is followed by GATv2Conv layers. Each GAT layer (except the last) is followed by ReLU activation. The final layer omits the activation function to preserve feature information.

- GCN encoder: This module consists of a stack of GCNConv layers. Similar to the GAT encoder, intermediate layers employ ReLU activation, while the final layer is linear.

**Quantum Evaluator Architecture ($f_\phi$).** The quantum evaluator employs a multi-view GNN architecture to fuse heterogeneous inputs. It comprises three parallel encoders and a prediction head, as detailed in Table 6.

**Joint Generator Architecture ($g_\theta$).** The joint generator follows a sequential "partition-first, parameter-second" design. Specifications are summarized in Table 7.

# B. Experimental Settings

## B.1. Hyperparameters

**Implementation and Environment.** We implemented Neural QAOA$^2$ using PyTorch and PyTorch Geometric. All experiments were conducted on a Linux workstation equipped with an Intel Xeon Platinum 8380 CPU (256 GB RAM) and four NVIDIA A30 GPUs (24 GB VRAM each). To ensure reproducibility, we fixed the random seed to 42 for Python, NumPy, and PyTorch backends.

**Optimization Strategy.** For all trainable components (quantum evaluator, joint generator), we employed the AdamW optimizer (Loshchilov & Hutter, 2019), which decouples weight decay from gradient updates to provide better generalization. Learning rates were dynamically adjusted using the ReduceLROnPlateau scheduler, which decays the learning rate when the validation loss stops improving.

**Training and Inference Phases.** Our optimization protocol consists of two offline training stages followed by online inference:

- Quantum evaluator training. The quantum evaluator $f_\phi$ was trained for 100 epochs with a batch size of 32. We used an initial learning rate of $1 \times 10^{-3}$ and a weight decay of $5 \times 10^{-4}$. The scheduler was configured with a factor of 0.5 and patience of 3 epochs.

- Joint generator pre-training. The joint generator $g_\theta$ was trained for 1500 epochs with a batch size of 16. We used an initial learning rate of $4 \times 10^{-3}$ and a weight decay of $5 \times 10^{-4}$. The scheduler used a factor of 0.8 with a patience of 100 steps.

- Test-time adaptation. During inference, for each test instance, we fine-tuned the generator parameters for 1000 steps (gradient updates) to maximize the predicted performance ratio. We utilized the same AdamW optimizer with a learning rate of $1 \times 10^{-3}$ and a scheduler (factor=0.8, patience=100, min_lr=$1 \times 10^{-4}$).

A comprehensive summary of all hyperparameters is provided in Table 8.

*Table 8.* **Hyperparameter Settings.** Summary of configurations for training and inference.

| Hyperparameter | Value |
|---|---|
| *Global Settings* | |
| Qubit Constraint (max_nodes) | 10 |
| QAOA Circuit Depth ($p$) | 1 |
| Random Seed | 42 |
| *Quantum Evaluator ($f_\phi$)* | |
| Batch Size | 32 |
| Optimizer | AdamW (Loshchilov & Hutter, 2019) |
| Learning Rate | $1 \times 10^{-3}$ |
| Weight Decay | $5 \times 10^{-4}$ |
| Training Epochs | 100 |
| LR Scheduler | ReduceLROnPlateau (Factor 0.5) |
| *Joint Generator ($g_\theta$)* | |
| Batch Size | 16 |
| Optimizer | AdamW (Loshchilov & Hutter, 2019) |
| Learning Rate | $4 \times 10^{-3}$ |
| Weight Decay | $5 \times 10^{-4}$ |
| Training Epochs | 1500 |
| LR Scheduler | ReduceLROnPlateau (Factor 0.8) |
| *Test-Time Adaptation (Inference)* | |
| Fine-tuning Steps | 1000 |
| Optimizer | AdamW (Loshchilov & Hutter, 2019) |
| Initial Learning Rate | $1 \times 10^{-3}$ |
| Minimal Learning Rate | $1 \times 10^{-4}$ |

### B.2. Dataset Details

**Source Benchmarks.** We evaluate Neural QAOA² on a public benchmark library. This library provides a diverse collection of problem instances with varying densities and topologies. Specifically, we utilize the following subsets:

- Training and validation: constructed from datasets *B*, *BE*, and *W* ($N \in [51, 501]$), representing standard QUBO and MaxCut problems. To ensure a reproducible evaluation, we adopt a deterministic splitting strategy based on instance indices: instances with IDs ending in 1 or 2 are reserved for testing (constituting 20%, or 50 graphs), while the remaining instances serve as the training and validation set (80%, or 200 graphs).

- Out-of-distribution testing: includes *GKA* (distinct QUBO distributions) and *L* (Ising spin glass with 1D/2D/3D lattice structures) to evaluate zero-shot generalization.

- Large-scale testing: includes *HR* and *BMZ* datasets ($N$ up to 1000) to assess performance on scales significantly larger than those seen during training.

**Data Preprocessing and Feature Engineering.** Raw graphs are converted into PyTorch Geometric `Data` objects. To ensure numerical stability and generalization across graph sizes, we perform the following preprocessing steps:

1. Node feature extraction: As detailed in Appendix A.3, we extract 5 structural features for each node. These features are standardized (zero mean, unit variance) within each graph instance to handle variations in graph scale.

2. Edge weight normalization: Edge weights are normalized by the maximum absolute weight in the graph, ensuring $w_{ij} \in [-1, 1]$.

**Construction of the Offline Dataset** ($\mathcal{D}_{\mathbf{offline}}$). To pre-train the quantum evaluator $f_\phi$, we constructed a labeled dataset consisting of 52587 triplets $(G_i, \mathbf{S}_i, \mathbf{P}_i)$ and their corresponding performance labels $\rho_i$. Crucially, this process relies solely on accessible samples and does not require computationally expensive optimal partitions $\mathbf{S}^*$ or parameters $\mathbf{P}^*$. The construction protocol is as follows:

1. Partition sampling ($\mathbf{S}_i$): To prevent the evaluator from overfitting to a specific partitioning logic, we employed a diverse sampling strategy. For each graph $G_i$, we generated partitions using a mixture of random, modularity, boundary, and KL heuristics. Each heuristic was executed independently 70 times per graph, with duplicate partitions removed to ensure diversity.

2. Parameter sampling ($\mathbf{P}_i$): QAOA parameters $(\boldsymbol{\gamma}, \boldsymbol{\beta})$ were sampled uniformly from $[0, 2\pi)$ to ensure the evaluator learns the landscape across the full parameter space.

3. Ground-truth label generation ($\rho_i$): The ground-truth performance labels were obtained by executing the QAOA² simulation using the configuration $(G_i, \mathbf{S}_i, \mathbf{P}_i)$. The entire offline data collection process was completed within approximately two days.

### B.3. Baseline Implementation

To benchmark the effectiveness of Neural QAOA², we compare it against four representative partitioning strategies. All baselines are implemented using the standard `NetworkX` library in Python, adapted to satisfy the hard constraint of maximum subgraph size (max_nodes).

**Random Partition.** Given a graph $G$ with $N$ nodes and a capacity constraint max_nodes, the algorithm randomly samples max_nodes vertices without replacement to form a subgraph, repeating this process until all vertices are assigned to $k = \lceil N/\text{max\_nodes} \rceil$ partitions.

As noted in previous studies (Zhou et al., 2023), random partitioning performs adequately on dense graphs where edges are uniformly distributed. However, on sparse graphs, it risks generating subgraphs with sparse internal connectivity and heavy inter-partition cuts, leading to suboptimal QAOA² performance.

**Greedy Modularity Maximization.** Proposed by Clauset et al. (Clauset et al., 2004), this method seeks to maximize the modularity ($Q$), a metric quantifying the strength of division of a network into modules. The algorithm follows a greedy agglomerative strategy, starting with each node in its own community and iteratively merging pairs that result in the maximal increase in $Q$. The modularity $Q$ is defined as:

$$Q = \frac{1}{2m} \sum_{i,j} \left[ \mathbf{A}_{ij} - \frac{d_i d_j}{2m} \right] \delta(c_i, c_j), \tag{22}$$

where $\mathbf{A}_{ij}$ is the element of the adjacency matrix, $d_i$ and $d_j$ denote the degrees of nodes $i$ and $j$, $m = \frac{1}{2} \sum_{i,j} \mathbf{A}_{ij}$ is the total edge weight, and $\delta(c_i, c_j)$ is the Kronecker delta function which equals 1 if nodes $i, j$ belong to the same community $c$ and 0 otherwise.

By maximizing intra-community edges relative to a null model, this heuristic aligns well with the divide-and-conquer objective of minimizing cuts between subgraphs, making it particularly effective for graphs with inherent community structures.

**Boundary Vertices Minimization.** Following the methodology of Guerreschi (2021), this baseline explicitly targets the reduction of boundary nodes, which are defined as nodes having at least one neighbor in a different partition. The implementation proceeds in two stages:

1. Initialization: A preliminary partition is generated using standard community detection (e.g., Louvain algorithm).

2. Refinement: A dedicated optimization routine iteratively moves vertices between partitions to minimize the count of boundary nodes while respecting the size constraint.

**Kernighan-Lin (KL) Algorithm.** The KL algorithm (Kernighan & Lin, 1970) is a classic heuristic for graph bisection that minimizes the edge cut weight. Starting from an initial random bisection, the algorithm iteratively swaps pairs of nodes between subsets to reduce the cut size until no further improvement is possible.

To satisfy our multi-partition requirement ($k > 2$) and size constraints, we apply the algorithm recursively. This ensures that the resulting subgraphs are disjoint and balanced, strictly adhering to the max_nodes limit.

## B.4. QAOA² Recursion and Quantum Simulation Protocol

To explicitly clarify the implementation details of the QAOA² recursion, we provide a formal breakdown of the recursion depth and the number of subproblem calls. The recursion structure is deterministically governed by the problem size $N$ and the hardware qubit constraint max_nodes (set to 10 in our main experiments).

**Recursion Logic.** Let $G^{(l)}$ denote the graph at recursion level $l$, with size $N^{(l)}$. As conceptually illustrated in Figure 2, the workflow consists of the following steps:

1. Partitioning: If $N^{(l)} > $ max_nodes, the graph is partitioned into $k^{(l)}$ subgraphs, satisfying $k^{(l)} \geq \lceil N^{(l)}/\text{max\_nodes} \rceil$.

2. Subproblem solving: Each of the $k^{(l)}$ subgraphs is solved independently via QAOA. This constitutes $k^{(l)}$ quantum circuit simulation calls at level $l$.

3. Reformulating (recursion step): The solutions to these subgraphs determine the edge weights of a new reformulated graph $G^{(l+1)}$ which has $N^{(l+1)} = k^{(l)}$ nodes.

4. Termination: This process repeats until the merged graph fits within the device, i.e., $N^{(L)} \leq$ max_nodes. The final graph is solved directly.

**Quantitative Analysis of Calls.** For a typical large-scale instance in our dataset (e.g., $N = 500$) with max_nodes $= 10$:

1. Level 0 (original graph): $N^{(0)} = 500$. Partitioned into $k^{(0)} = 50$ subgraphs. (50 calls).

2. Level 1 (first reformulated graph): The reformulated graph has size $N^{(1)} = 50$. Since $50 > 10$, it is partitioned into $k^{(1)} = 5$ subgraphs. (5 calls).

3. Level 2 (final reformulated graph): The new reformulated graph has size $N^{(2)} = 5$. Since $5 \leq 10$, recursion terminates, and this graph is solved directly. (1 call).

Total depth is 2 (excluding the base layer), and the total number of QAOA subproblem calls is $50 + 5 + 1 = 56$. This hierarchical structure results in a recursion depth of $\mathcal{O}(\log_{\text{max\_nodes}} N)$. Consequently, the total number of subproblem calls scales linearly as $\mathcal{O}(N)$, ensuring that the total quantum simulation cost remains tractable even for large $N$. All baselines and our Neural QAOA² follow this identical recursion protocol to ensure fair wall-clock time comparisons.

**QAOA Implementation Details.** To ensure strict computational fairness and reproducibility, the optimization routine for QAOA simulations is standardized across all methods. For each subproblem, we execute a standard QAOA routine implemented using PennyLane (Bergholm et al., 2018) with the `lightning.gpu` backend to accelerate simulation. The detailed configuration is as follows:

- Circuit optimization: We utilize the standard gradient descent optimizer with a fixed learning rate (step size) of 0.01. The optimization budget is fixed at 20 steps for all subproblems in the main experiments. Gradients are computed via the adjoint differentiation method (Jones & Gacon, 2020), which provides exact gradients with constant memory overhead, ensuring efficient training on GPU.

- Sampling protocol: After parameter optimization, we perform sampling with 1000 shots.

*Table 9.* **Comprehensive Evaluation on** $p = 2, 3$ **with Advanced Parameter Initialization Strategies.** We compare Neural QAOA² (Ours) against the top two partition heuristics, QAOA² (Modularity, Zhou et al., 2023; Clauset et al., 2004) and QAOA² (Boundary, Guer-reschi, 2021), equipped with advanced parameter initialization strategies. Since Neural QAOA² inherently generates partitions and initial parameters, it is presented as a single end-to-end solution. Parameter initialization strategies are categorized into (a) topology-blind: **RI** (random initialization); (b) physics-inspired: **TQA** (Trotterized quantum annealing, Sack & Serbyn, 2021); **INT** (INTERP, Zhou et al., 2020); **FOU** (FOURIER, Zhou et al., 2020); (c) learning-based: **QIBPI** (quantum instance-based parameter initialization, Katial et al., 2025). Cell layout: mean $\bar{\rho}$ (top), $\pm$ std (middle), and rank (bottom, calculated among all 11 columns). **Bold** indicates the best performance.

| Dataset (#) | QAOA² (Modularity) | | | | | QAOA² (Boundary) | | | | | Neural QAOA² (Ours) |
|---|---|---|---|---|---|---|---|---|---|---|---|
| | RI | TQA | INT | FOU | QIBPI | RI | TQA | INT | FOU | QIBPI | GEN |
| *Circuit Depth: $p = 2$* | | | | | | | | | | | |
| *B* (8) | 0.8333 | 0.8323 | 0.8314 | 0.8314 | 0.8350 | 0.8255 | 0.8263 | 0.8252 | 0.8252 | 0.8275 | **0.8420** |
| | ± 0.0323 | ± 0.0272 | ± 0.0291 | ± 0.0291 | ± 0.0281 | ± 0.0272 | ± 0.0274 | ± 0.0317 | ± 0.0317 | ± 0.0301 | ± 0.0209 |
| | (5.38) | (6.25) | (5.38) | (5.38) | (5.13) | (8.50) | (6.50) | (6.13) | (6.13) | (6.38) | **(2.88)** |
| *BE* (16) | 0.8707 | 0.8705 | 0.8709 | 0.8709 | 0.8697 | 0.8718 | 0.8716 | 0.8712 | 0.8712 | 0.8716 | **0.8802** |
| | ± 0.0327 | ± 0.0333 | ± 0.0338 | ± 0.0338 | ± 0.0353 | ± 0.0360 | ± 0.0348 | ± 0.0341 | ± 0.0341 | ± 0.0352 | ± 0.0319 |
| | (6.81) | (6.69) | (6.63) | (6.63) | (7.50) | (5.25) | (6.06) | (5.44) | (5.44) | (5.81) | **(1.75)** |
| *W* (26) | 0.9127 | 0.9141 | 0.9145 | 0.9145 | 0.9139 | 0.9125 | 0.9112 | 0.9115 | 0.9115 | 0.9108 | **0.9160** |
| | ± 0.0304 | ± 0.0297 | ± 0.0295 | ± 0.0295 | ± 0.0300 | ± 0.0331 | ± 0.0338 | ± 0.0348 | ± 0.0348 | ± 0.0344 | ± 0.0270 |
| | (6.85) | (5.23) | **(4.54)** | **(4.54)** | (5.42) | (6.38) | (6.92) | (6.04) | (6.04) | (6.65) | (5.08) |
| Overall (50) | 0.8866 | 0.8870 | 0.8873 | 0.8873 | 0.8871 | 0.8855 | 0.8849 | 0.8848 | 0.8848 | 0.8849 | **0.8927** |
| | ± 0.0434 | ± 0.0433 | ± 0.0439 | ± 0.0439 | ± 0.0436 | ± 0.0460 | ± 0.0455 | ± 0.0465 | ± 0.0465 | ± 0.0457 | ± 0.0390 |
| | (6.60) | (5.86) | (5.34) | (5.34) | (6.04) | (6.36) | (6.58) | (5.86) | (5.86) | (6.34) | **(3.66)** |
| *Circuit Depth: $p = 3$* | | | | | | | | | | | |
| *B* (8) | 0.8359 | 0.8353 | 0.8352 | 0.8338 | 0.8383 | 0.8227 | 0.8289 | 0.8264 | 0.8276 | 0.8243 | **0.8436** |
| | ± 0.0287 | ± 0.0300 | ± 0.0279 | ± 0.0287 | ± 0.0319 | ± 0.0307 | ± 0.0373 | ± 0.0278 | ± 0.0334 | ± 0.0322 | ± 0.0247 |
| | (5.88) | (6.50) | (5.38) | (7.25) | (4.50) | (7.88) | (6.75) | (5.88) | (5.38) | (8.13) | **(2.50)** |
| *BE* (16) | 0.8700 | 0.8686 | 0.8708 | 0.8708 | 0.8693 | 0.8714 | 0.8718 | 0.8713 | 0.8708 | 0.8721 | **0.8823** |
| | ± 0.0342 | ± 0.0354 | ± 0.0329 | ± 0.0340 | ± 0.0342 | ± 0.0365 | ± 0.0374 | ± 0.0358 | ± 0.0347 | ± 0.0360 | ± 0.0315 |
| | (6.50) | (8.38) | (6.63) | (6.44) | (8.13) | (5.75) | (5.56) | (5.69) | (6.13) | (5.06) | **(1.75)** |
| *W* (26) | 0.9132 | 0.9130 | 0.9130 | 0.9140 | 0.9132 | 0.9106 | 0.9107 | 0.9116 | 0.9122 | 0.9127 | **0.9158** |
| | ± 0.0306 | ± 0.0322 | ± 0.0305 | ± 0.0299 | ± 0.0311 | ± 0.0339 | ± 0.0341 | ± 0.0347 | ± 0.0342 | ± 0.0332 | ± 0.0272 |
| | (5.46) | (5.69) | (5.96) | **(5.04)** | (5.46) | (7.31) | (7.08) | (6.23) | (6.31) | (5.92) | (5.15) |
| Overall (50) | 0.8870 | 0.8864 | 0.8870 | 0.8873 | 0.8872 | 0.8840 | 0.8852 | 0.8850 | 0.8854 | 0.8856 | **0.8936** |
| | ± 0.0431 | ± 0.0444 | ± 0.0426 | ± 0.0434 | ± 0.0433 | ± 0.0468 | ± 0.0466 | ± 0.0462 | ± 0.0463 | ± 0.0468 | ± 0.0387 |
| | (5.86) | (6.68) | (6.08) | (5.84) | (6.16) | (6.90) | (6.54) | (6.00) | (6.10) | (6.00) | **(3.64)** |

# C. Additional Experimental Results

## C.1. Advanced Parameter Initialization Strategies and Deeper Quantum Circuits

While the main experiments adopt random initialization (RI), we strengthen the two most competitive heuristic baselines (modularity and boundary) by equipping them with four advanced QAOA parameter initialization strategies: the physics-inspired TQA (Sack & Serbyn, 2021), INTERP (Zhou et al., 2020), FOURIER (Zhou et al., 2020), and the learning-based QIBPI (Katial et al., 2025). In addition, we increase the circuit depth to $p = 2$ and $p = 3$ to assess performance scaling under deeper quantum circuits.

As summarized in Table 9, advanced initialization strategies improve heuristic baselines over random initialization in more than half of the evaluated combinations. However, they remain insufficient to close the performance gap to Neural QAOA². For example, on the *B* dataset with $p = 2$, Neural QAOA² achieves an average rank of 2.88, substantially outperforming the strongest baseline configuration (modularity+QIBPI, rank 5.13). These results indicate that heuristic partitioning introduces a fundamental structural bottleneck, which cannot be overcome by parameter optimization alone, thereby underscoring the necessity of jointly learning partitioning and parameter initialization.

We further evaluate the adaptability of our approach to deeper circuits, despite both the quantum evaluator and the joint generator being trained exclusively on $p = 1$ QAOA² simulation data. To adapt to deeper circuits ($p = 2, 3$), we employ the physics-inspired INTERP strategy (Zhou et al., 2020) to extrapolate the synthesized depth-1 parameters to higher depths. Empirically, this extrapolation incurs no noticeable degradation in performance: Neural QAOA² consistently outperforms all ten competing baseline combinations at both $p = 2$ and $p = 3$. This demonstrates that the learned partitions, together with the generated parameter initialization, generalize effectively to increased circuit depth without requiring retraining on deeper circuits.

## C.2. Generalization to Hardware Constraints

In real-world quantum computing scenarios, the number of available physical qubits varies across different devices and platforms. A practical neural solver must adapt to these hardware constraints without requiring resource-intensive retraining. To evaluate this capability, we test Neural QAOA² on the test instances from Section 5.2 while varying the subproblem size limit max_nodes $\in [5, 15]$ (representing available physical qubits).

Figure 6 reports the results. Generally, larger subproblems allow for more extensive local optimization, naturally improving the performance ratio across all methods. Neural QAOA² consistently achieves the best average rank across all settings. Even at smaller qubit scales (max_nodes $< 10$) that are tighter than the training configuration (max_nodes $= 10$), our method establishes a substantial performance gap against baselines. This confirms that GEN enables zero-shot generalization to quantum devices with diverse qubit capacities.

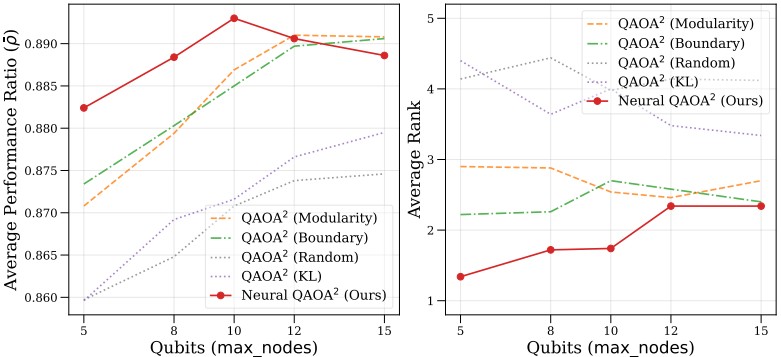

*Figure 6.* **Hardware Resources Generalization.** Performance comparison under varying subproblem size limits (max_nodes $\in [5, 15]$), representing available physical qubits. Neural QAOA² demonstrates strong resource generalization, consistently achieving the best average rank (right) and competitive mean $\bar{\rho}$ (left) without requiring retraining.

## C.3. Generalization to Unseen Graph Topologies

To investigate whether a model trained on Erdős–Rényi (ER) graphs can generalize to other topologies, such as Power-Law or Regular graphs, we additionally evaluate an ER-trained model on unseen Power-Law and Regular graphs at $N = 100$. The experimental settings remain identical to those detailed in Section 5.2.

The average performance ratios are summarized in Table 10. These results suggest that the learned mapping does not simply overfit to the ER training distribution. Instead, it achieves the strongest performance on Power-Law graphs and remains competitive on Regular graphs, performing very closely to the best heuristic baseline.

*Table 10.* **Average Performance Ratios on Unseen Graph Topologies.** The model is trained on ER graphs and evaluated on ER, Power-Law, and Regular graphs at $N = 100$. The numbers in parentheses indicate the number of test instances for each distribution.

| Method | ER (50, In-dist) | Power-Law (25) | Regular (25) |
|---|---|---|---|
| QAOA² (Random) | 0.8262 | 0.7715 | 0.7492 |
| QAOA² (Modularity) | 0.8645 | 0.8548 | **0.8643** |
| QAOA² (Boundary) | 0.8580 | 0.8385 | 0.8514 |
| QAOA² (KL) | 0.8211 | 0.7687 | 0.7485 |
| **Neural QAOA² (Ours)** | **0.8717** | **0.8566** | 0.8628 |

## C.4. Evaluator Calibration and Reliability Analysis

To address concerns regarding the potential "over-optimization" of the learned quantum evaluator $f_\phi$ during test-time adaptation (TTA), we conduct a calibration study on configurations generated by the joint generator $g_\theta$. Figure 7 illustrates the correlation between the predicted performance ratio ($\hat{\rho}$) and the ground-truth performance ratio ($\rho$) obtained via quantum simulation.

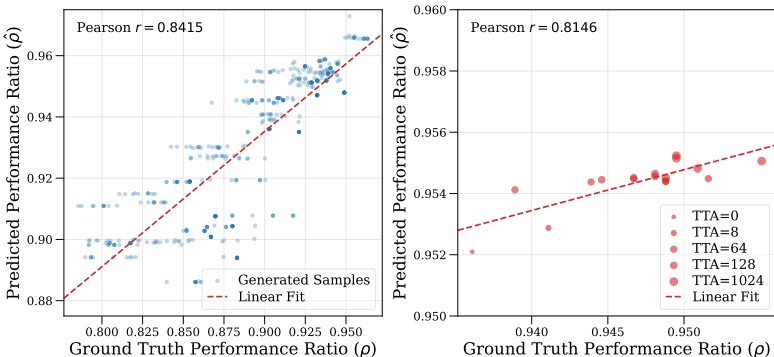

*Figure 7.* **Evaluator Calibration on Generator-Produced Samples.** Left: global correlation between predicted ($\hat{\rho}$) and ground-truth ($\rho$) performance ratios. The scatter plot comprises 500 configurations $(\mathbf{S}, \mathbf{P})$ collected from 10 independent runs of Neural QAOA² (with 64 fine-tuning steps) across the 50 test instances used in Section 5.2. Right: analysis on the representative instance *g05_100.1* (analyzed in Section 5.7). Data points represent configurations generated from 5 independent runs at various stages of test-time adaptation (TTA). The consistently high Pearson correlation ($r > 0.81$) demonstrates that the evaluator remains well-calibrated both globally and locally during the optimization trajectory, effectively mitigating the risk of distribution shift in high-performance regions.

**Robustness to Distribution Shift.** As illustrated in Figure 7 (Left), to rigorously assess the global reliability of our surrogate, we aggregated 500 generated configurations $(\mathbf{S}, \mathbf{P})$ from 10 independent inference runs across the 50 test instances used in Section 5.2. Specifically, these samples were collected after 64 steps of fine-tuning, a stage where the generator is expected to produce high-quality joint partition and parameter configurations that deviate from the initial heuristic-based training distribution. Despite this potential distribution shift, the evaluator maintains a high Pearson correlation of $r = 0.8415$. This strong linear alignment confirms that the evaluator's predictions remain tightly coupled with the ground truth even in the high-performance regions favored by the learned generator, strongly mitigating concerns about "gaming" a fragile surrogate.

**Accuracy during Test-Time Adaptation.** To verify local fidelity, Figure 7 (Right) visualizes the configurations sampled from 5 independent optimization runs on the representative instance *g05_100.1*. We explicitly track the correlation at various stages of TTA. Even at 1024 steps of fine-tuning, where the generator explores the unseen high-performance region beyond the initial heuristic distribution, the evaluator maintains a strong correlation of $r = 0.8146$. This alignment alleviates concerns that the generator might game the surrogate objective, and indicates that the evaluator provides a reliable gradient signal that effectively guides the search for improved quantum partitions and parameters.

### C.5. Robustness to Evaluator Prediction Errors

To investigate the robustness of our generator to prediction errors in the evaluator, we assess the average performance ratio ($\bar{\rho}$) using evaluator checkpoints from earlier training epochs. These earlier checkpoints exhibit higher prediction errors, measured by the validation mean squared error (MSE).

The results are summarized in Table 11. Differences in performance compared to the fully converged model (Epoch 100) are evaluated for statistical significance using a one-sided paired t-test ($p < 0.01$). As shown, a statistically significant degradation in $\bar{\rho}$ is observed only when the prediction error increases substantially (e.g., by approximately $1.67\times$ at Epoch 1). These results suggest that the proposed method remains reasonably tolerant to moderate inaccuracies in the surrogate evaluator.

*Table 11.* **Robustness Analysis of the Generator Against Evaluator Prediction Errors.** Evaluator checkpoints from earlier training epochs (higher validation MSE) are used to guide the generator. An asterisk (*) denotes a statistically significant performance drop compared to the fully converged model (Epoch 100), determined via a one-sided paired t-test ($p < 0.01$).

| Epoch | Prediction Error ($\times 10^{-4}$) | Mean $\bar{\rho}$ |
|-------|-------------------------------------|-------------------|
| 100   | 1.92                                | 0.8930            |
| 20    | 1.99                                | 0.8923            |
| 2     | 2.82                                | 0.8903            |
| 1     | 3.21                                | 0.8879*           |

## C.6. Computational Cost versus Problem Size

The overall computational cost comprises two components: the offline training cost and the online inference cost.

**Offline Training Cost.** We decompose the offline training cost into two distinct phases: offline dataset construction and model training. Building the offline dataset takes approximately two days of wall-clock time using four NVIDIA A30 GPUs. As the problem size $N$ increases, this construction overhead grows roughly linearly within the tested regime (see Appendix B.2 and Appendix B.4). For the model training, the evaluator and generator require approximately 2.65 hours (for 100 epochs) and 2.12 hours (for 1,500 epochs), respectively. To analyze how this training cost scales with $N$, we record the wall-clock time per training batch, as detailed in Table 12. Notably, when $N$ increases by a factor of 10 (from 51 to 501), the batch training time increases by only $2\times$ for the evaluator and $3.3\times$ for the generator, which is sub-linear scaling.

*Table 12.* Average wall-clock time per training batch for the evaluator and generator across varying problem sizes ($N$). The batch sizes are fixed at 32 for the evaluator and 16 for the generator. All measurements were conducted on NVIDIA A30 GPUs.

| Problem Size ($N$) | Evaluator Time (ms) | Generator Time (ms) |
|---|---|---|
| 51 | 67 | 123 |
| 101 | 71 | 144 |
| 251 | 72 | 334 |
| 501 | 134 | 406 |

Overall, as $N$ increases, the dataset construction cost grows roughly linearly, while the model training time scales sub-linearly. Therefore, the total offline training cost remains scalable with respect to the problem size $N$.

**Online Inference Cost.** We evaluate the end-to-end inference time under a fixed budget of 64 test-time adaptation steps. As detailed in Table 13, the runtime increases substantially with the problem size, exhibiting a near-linear growth trend within the tested regime. The dominant computational overhead stems from the repeated forward and backward adaptation passes, combined with the quantum circuit simulation calls inherent to the QAOA² pipeline.

*Table 13.* Online inference time across varying problem sizes ($N$) under a fixed 64-step adaptation budget.

| Problem Size ($N$) | Total Time (s) |
|---|---|
| 51 | 50.3 |
| 101 | 87.0 |
| 251 | 174.3 |
| 501 | 435.6 |

## C.7. Extended Comparisons with Classical and Hybrid Solvers

To provide a more comprehensive evaluation, we extend our comparisons to include a fine-tuned version of the hybrid PI-GNN solver (Schuetz et al., 2022) and the classical Goemans-Williamson (GW) algorithm (Goemans & Williamson, 1995). For PI-GNN, we conduct a grid search over two key hyperparameters (learning rates and dropout probabilities) based on the official repository[3], retaining default values for the remainder. The detailed configurations for PI-GNN and GW are provided in Table 15 and Table 16, respectively.

The performance comparison is summarized in Table 14, with results reported as the mean performance ratio alongside the standard deviation to illustrate the variance across instances. Additionally, Figure 8 provides boxplots to better visualize the performance distribution. As observed, the results are mixed and comparable; no single method dominates others across all datasets.

We emphasize that the contribution of this work is not to establish empirical supremacy over classical heuristics, which remains a long-term challenge for quantum combinatorial optimization (Gemeinhardt et al., 2023). Rather, it is introducing a differentiable, end-to-end learning paradigm to overcome the partitioning and initialization bottlenecks within the QAOA² framework. Advancing the scalability of QAOA under realistic constraints is valuable in this area.

---

[3]https://github.com/amazon-science/co-with-gnns-example

*Table 14.* Performance comparison of our method against the fine-tuned PI-GNN and the classical Goemans-Williamson (GW) heuristic. Results are reported as the mean performance ratio ± standard deviation.

| Dataset (# Instances) | PI-GNN | GW | **Ours** |
|---|---|---|---|
| *B* (8) | $0.9416 \pm 0.0157$ | $0.8141 \pm 0.0240$ | $0.8417 \pm 0.0256$ |
| *BE* (16) | $0.9311 \pm 0.0098$ | $0.8664 \pm 0.0209$ | $0.8824 \pm 0.0304$ |
| *W* (26) | $0.9106 \pm 0.0426$ | $0.9473 \pm 0.0117$ | $0.9153 \pm 0.0280$ |

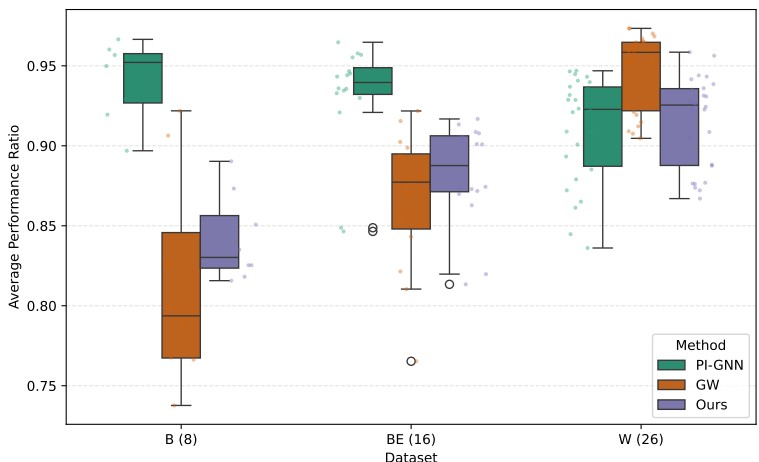

*Figure 8.* Boxplots of the average performance ratio across three datasets: *B* (8), *BE* (16), and *W* (26), comparing PI-GNN, GW, and our method. For each dataset, three side-by-side boxplots are presented, with colors indicating the respective methods and the y-axis reporting the average performance ratio (higher is better). Each boxplot is constructed from all instance-level results of a given dataset-method pair: the center line marks the median, the box bounds are the first and third quartiles (Q1 and Q3), and the box height indicates the interquartile range (IQR). The whiskers extend to the most extreme observations within [Q1 - 1.5IQR, Q3 + 1.5IQR], while points outside this range are shown as outliers.

*Table 15.* Detailed hyperparameter settings for the fine-tuned PI-GNN baseline. These configurations correspond to the optimal setup identified in our grid search, adhering to the recommendations from the official repository.

| Parameter | Value | Description |
|---|---|---|
| learning_rate | $10^{-3}$ | Adam learning rate; controls the optimization step size. |
| number_epochs | 100,000 | Maximum training epochs per run. |
| prob_threshold | 0.5 | Threshold for binarizing node probabilities into partition bits. |
| tolerance | $10^{-4}$ | Early-stopping loss-delta tolerance. |
| patience | 100 | Early-stopping patience (consecutive epochs). |
| dropout | 0.5 | Dropout probability for hidden layers (regularization). |
| n_layers | 2 | Number of GNN hidden propagation layers. |

*Table 16.* SDP solver configurations and randomized rounding parameters for the Goemans-Williamson (GW) baseline.

| Parameter | Value | Description |
|---|---|---|
| solver | SCS | SDP solver used for the GW relaxation. |
| max_iters | 25 | Maximum SCS iterations. |
| eps | $10^{-4}$ | SCS convergence tolerance. |
| alpha | 1.8 | SCS relaxation parameter. |
| scale | 5.0 | SCS data scaling parameter. |
| random_hyperplanes | 5 | Number of randomized hyperplane rounding trials. |

*Table 17.* Generalization performance across varying problem sizes ($N$) when training on different fractions of the offline dataset. A statistically significant performance drop compared to the 100% model ($p < 0.05$, one-sided paired t-test) is denoted by "*".

| Train Fraction (Size) | $N = 51$ (p-value) | $N = 100$ (p-value) | $N = 501$ (p-value) | $N = 800$ (p-value) | $N = 1000$ (p-value) |
|---|---|---|---|---|---|
| 100% (42,069) | 0.9125 (N/A) | 0.9059 (N/A) | 0.8935 (N/A) | 0.8670 (N/A) | 0.7244 (N/A) |
| 80% (33,655) | - | - | 0.8925 ($7.26 \times 10^{-2}$) | 0.8663 ($1.10 \times 10^{-1}$) | 0.7229 ($9.60 \times 10^{-2}$) |
| 60% (25,241) | 0.9110 ($2.35 \times 10^{-1}$) | 0.9042 ($1.66 \times 10^{-1}$) | 0.8923* ($4.41 \times 10^{-2}$) | 0.8655* ($2.07 \times 10^{-2}$) | 0.7221* ($2.68 \times 10^{-2}$) |
| 40% (16,827) | 0.9087 ($7.12 \times 10^{-2}$) | 0.8998* ($4.41 \times 10^{-2}$) | 0.8919* ($2.16 \times 10^{-2}$) | 0.8652* ($9.07 \times 10^{-4}$) | 0.7212* ($3.10 \times 10^{-3}$) |
| 20% (8,314) | 0.9030* ($4.35 \times 10^{-2}$) | 0.9007* ($3.71 \times 10^{-2}$) | - | - | - |

## C.8. Sample-efficiency Ablation Study

To further evaluate the sample-efficiency of the proposed method under reduced offline supervision, we additionally conduct an ablation study by training the evaluator on nested subsets containing 20%, 40%, 60%, 80%, and 100% of the original offline dataset. Crucially, the validation and test sets remain strictly identical across all settings to ensure fair comparison. The performance across different problem sizes $N \in \{51, 100, 501, 800, 1000\}$ is reported in Table 17. Statistical significance against the 100% training setting is evaluated using a one-sided paired t-test, where a significant performance degradation ($p < 0.05$) is marked with "*". We define "sufficient generalization" as achieving no statistically significant degradation compared to the full-dataset training setting.

Overall, the data requirement exhibits a roughly monotonic scaling trend: while only 40% of the data is sufficient for generalizing to $N = 51$, the requirement increases to 60% for $N = 100$. For large-scale generalization ($N = 501, 800, 1000$), the model demands denser coverage of the training space, requiring at least 80% (approximately 33k samples) to avoid statistically significant degradation. This suggests that generalizing to larger problem sizes necessitates a larger offline dataset, though the requirement empirically appears to saturate around the 80% threshold for the tested scales.

# D. Theoretical Analysis

In this section, we provide the detailed proof for the boundedness of our proposed performance ratio metric $\rho$, as stated in Proposition 4.1.

## D.1. Proof of Proposition 4.1

*Proof.* **Restatement of Proposition:** *For any graph $G$, the metric $\rho$, evaluated on the output of the QAOA², is theoretically bounded in* $[0.5, 1.0]$.

**Derivation:** Recall the definition of the metric:

$$\rho = \frac{\text{Cut}(G) - \text{Neg}(G)}{\text{OPT}(G) - \text{Neg}(G)}. \tag{23}$$

Let $W_{\text{total}} = \sum_{(u,v) \in E} |w_{uv}|$ be the sum of absolute weights of all edges, where we assume $W_{\text{total}} > 0$. We can decompose the total weight into positive and negative components: $W_{\text{total}} = \text{Pos}(G) + |\text{Neg}(G)|$. Note that $\text{Neg}(G)$ represents the sum of negative edge weights (a negative number), implying $-\text{Neg}(G) = |\text{Neg}(G)|$.

The numerator can be interpreted as the cut weight on a shifted spectrum where all edge weights are non-negative:

$$\text{Numerator} = \text{Cut}(G) + |\text{Neg}(G)|. \tag{24}$$

**Upper Bound ($\leq 1.0$):** Since $\text{Cut}(G) \leq \text{OPT}(G)$ holds by definition, and the denominator is positive, it trivially follows that:

$$\rho = \frac{\text{Cut}(G) - \text{Neg}(G)}{\text{OPT}(G) - \text{Neg}(G)} \leq 1.0. \tag{25}$$

**Lower Bound ($\geq 0.5$):** This lower bound is guaranteed by the theoretical properties of the QAOA² framework. To establish the lower bound, we invoke Theorem 2 from Zhou et al. (2023), which proves that the QAOA² framework theoretically guarantees that the output solution quality lower-bounded is by the expectation of a random cut. Specifically, the theorem

states:

$$\text{Cut}(G) \geq \frac{1}{2} \sum_{(u,v) \in E} w_{uv} = \frac{1}{2} \left( \text{Pos}(G) + \text{Neg}(G) \right). \tag{26}$$

The validity of Proposition 4.1 relies on this standard algorithmic property (i.e., the optimized circuit performs no worse than random assignment). Under this premise, we substitute the lower bound into the numerator of Eq. (23):

$$
\begin{aligned}
\text{Numerator} &= \text{Cut}(G) - \text{Neg}(G) \\
&\geq \frac{1}{2} \left( \text{Pos}(G) + \text{Neg}(G) \right) - \text{Neg}(G) \\
&= \frac{1}{2} \text{Pos}(G) - \frac{1}{2} \text{Neg}(G).
\end{aligned}
\tag{27}
$$

Since $-\text{Neg}(G) = |\text{Neg}(G)|$, the numerator satisfies:

$$\text{Numerator} \geq \frac{1}{2} \left( \text{Pos}(G) + |\text{Neg}(G)| \right) = \frac{1}{2} W_{\text{total}}. \tag{28}$$

For the denominator, we observe that the optimal cut value $\text{OPT}(G)$ is naturally upper-bounded by the sum of all positive edge weights (corresponding to the ideal scenario where all positive edges are cut and no negative edges are cut):

$$\text{OPT}(G) \leq \text{Pos}(G). \tag{29}$$

Consequently, the denominator is bounded by:

$$
\begin{aligned}
\text{Denominator} &= \text{OPT}(G) - \text{Neg}(G) \\
&\leq \text{Pos}(G) - \text{Neg}(G) \\
&= \text{Pos}(G) + |\text{Neg}(G)| = W_{\text{total}}.
\end{aligned}
\tag{30}
$$

Combining the bounds for the numerator and the denominator, we derive the lower bound for the performance ratio:

$$\rho = \frac{\text{Numerator}}{\text{Denominator}} \geq \frac{\frac{1}{2} W_{\text{total}}}{W_{\text{total}}} = 0.5. \tag{31}$$

Therefore, we conclude that $\rho \in [0.5, 1.0]$. $\qquad\square$

# E. Extended Related Work

**Scalable QAOA beyond D&C.** Beyond divide-and-conquer strategies, scalable QAOA has also been explored through alternative paradigms that reduce effective problem size. Representative approaches include recursive variable elimination, such as R-QAOA (Bravyi et al., 2020), and multilevel coarsening methods, such as MLQAOA (Bach et al., 2024), which construct hierarchical problem representations to enable execution on limited quantum resources.

**Learning-based Combinatorial Optimization.** Deep learning for combinatorial optimization has evolved from sequential reinforcement learning approaches (Khalil et al., 2017; Liu et al., 2023b), parallel unsupervised relaxation methods (Schuetz et al., 2022), to large language model-driven enhancements (Liu et al., 2024; Zhang et al., 2025; Jiang et al., 2025). Recent works (Ichikawa, 2024; Abate & Bianchi, 2025; Shen et al., 2025; Ichikawa & Arai, 2025; Lv et al., 2025) have demonstrated strong performance improvements in classical solvers. In the context of quantum optimization, an emerging line of research explores how deep learning can complement, rather than replace, quantum algorithms. Notably, prior studies have investigated supervised training of quantum neural networks (Ye et al., 2023; 2025) as well as learning-based customization of mixer unitary ansatz structures, such as MG-Net (Qian et al., 2024). Complementary to these efforts, we address limitations in scalable QAOA², utilizing neural networks to jointly generate partitions and initial circuit parameters.

# F. Additional Discussion

**Barren Plateaus.** Our method does not claim to solve barren plateaus in general. Our method is trained only at $p = 1$, so we do not optimize a high-depth circuit from scratch, thereby circumventing severe barren-plateau issues. For $p > 1$, we use

the standard parameter expansion strategy from prior work (Zhou et al., 2020) rather than retraining the model end-to-end in the larger parameter space. Thus, the current framework largely avoids the most difficult high-depth training regime rather than fundamentally resolving barren plateaus. Empirically, under this protocol we still observe competitive results at $p = 2, 3$ in Appendix C.1.

**Two-Stage Training.** We adopted the decoupled two-stage design deliberately to ensure optimization stability and tractability. While meta-learning approaches (e.g., MAML) could optimize the generator more explicitly for post-adaptation quantum performance rather than against a frozen surrogate alone, they would require differentiating through an inner adaptation loop, leading to substantially higher computational and memory cost, and potentially much noisier optimization in our quantum-in-the-loop setting. Therefore, optimizing against a frozen surrogate serves as a much more practical and stable first step.

**Surrogate Sensitivity.** To assess the sensitivity to surrogate inaccuracies, a critical question is whether downstream optimization degrades materially when the evaluator is imperfect. We examine this directly in two ways: First, during long test-time adaptation (TTA) trajectories (see Appendix C.4, Figure 7), the evaluator remains well-aligned with the true QAOA² performance even in newly explored regions. Specifically, it maintains a correlation of $0.8146$ after $1,024$ TTA steps, suggesting that the surrogate stays informative beyond the densest part of the offline training distribution. Second, we conduct ablation experiments intentionally using less accurate evaluators from earlier training epochs. The average performance ratio $\bar{\rho}$ exhibits only a mild degradation from $0.8930$ (using a fully converged evaluator) to $0.8923$ and $0.8903$ for earlier checkpoints, and drops significantly ($0.8879$) only when using the least accurate checkpoint (see Appendix C.5). Thus, while the objective is indeed indirect, the generator appears reasonably robust to moderate surrogate inaccuracies rather than highly fragile to them.

**Latency-Quality Trade-off.** We additionally analyze the tradeoff between fine-tuning cost and achievable solution quality across different problem sizes. We suggest evaluating this tradeoff through the lens of Pareto optimality. Specifically, the proposed method achieves Pareto dominance over existing QAOA² baselines in terms of solution quality versus computational cost. Under a fixed fine-tuning budget of 64 steps, our method already outperforms existing QAOA² baselines on average while maintaining comparable computational cost (see Section 5.5, Figure 4), indicating that the proposed approach is a Pareto optimal method. For large-scale problems, existing QAOA² methods often fail to obtain higher-quality solutions, while our method can reach these solutions if a larger budget is used. By doing so, we push the current Pareto frontier forward.

To further characterize the scaling behavior of test-time adaptation, we define the effective fine-tuning steps ($T_{\text{eff}}$) as the minimum number of fine-tuning steps required to recover 90% of the total improvement between the performance without fine-tuning steps and the empirical ceiling performance (measured at 1000 fine-tuning steps).

*Table 18.* Scaling behavior of test-time fine-tuning across different problem sizes. $T_{\text{eff}}$ denotes the minimum number of fine-tuning steps required to recover 90% of the achievable improvement, while $\tau$ denotes the average wall-clock time per fine-tuning step.

| Problem Size ($N$) | $T_{\text{eff}}$ | $\tau$ (ms) |
|---|---|---|
| 51 | 16 | 21 |
| 101 | 64 | 28 |
| 251 | 256 | 59 |
| 501 | 512 | 151 |

The results suggest that the adaptation budget required to capture most of the achievable gain increases with problem size in the tested regime. However, the scaling trend remains almost linearly in the large-scale regime (e.g., $T_{\text{eff}}$ doubles from 256 to 512 as $N$ increases from 251 to 501). Furthermore, the wall-clock overhead of each fine-tuning step is on the millisecond scale due to GPU acceleration. Table 18 reports the average wall-clock time per fine-tuning step across different problem sizes. These findings suggest that allocating larger fine-tuning budgets for large-scale instances is computationally practical.

