# OpenReview forum: "Neural QAOA$^2$: Differentiable Joint Graph Partitioning and Parameter Initialization for Quantum Combinatorial Optimization"
_ICML.cc/2026/Conference — ICML 2026 regular_

### Official Review · Reviewer_m98V · 2026-03-09

**Soundness:** 3
**Presentation:** 3
**Significance:** 2
**Originality:** 2
**Overall Recommendation:** 3
**Confidence:** 3

**Summary:**

This paper proposes Neural QAOA$^2$, an end-to-end differentiable framework to improve QAOA within the divide-and-conquer paradigm. The method jointly learns graph partitions and topology-aware initial parameters by integrating a generative evaluative network that utilizes a differentiable quantum evaluator. The goal is to better align partitioning with quantum optimization objectives and avoid topology-blind random parameter initialization. Noiseless simulations on QUBO, Ising, and MaxCut instances show improvements over heuristic partitioning and initialization strategies and suggest good generalization across different graph sizes and structures.

**Compliance With Llm Reviewing Policy:**

Affirmed.

**Key Questions For Authors:**

1. How does the computational cost of neural QAOA$^2$ scale with the problem size?
2. The generator is optimized against $f_\phi$ rather that the true QAOA$^2$. How robust is the method to prediction errors in this evaluator?

**Limitations:**

Yes.

**Strengths And Weaknesses:**

**Strengths**
1. The idea of using a learned surrogate evaluator and a generator to guide partitioning and parameter selection is conceptually interesting.
2. The numerical experiments on QUBO, Ising, and MaxCut instances show consistent improvements over heuristic partitioning and random initialization strategies within the QAOA$^2$ framework.
3. The numerical experiments suggest that the neural QAOA$^2$ can generalize across different graph distributions and sizes.

**Weaknesses**
1. The paper mainly compares against QAOA$^2$ baselines. Since neural QAOA$^2$ introduces additional learned components, it is not surprising that it improves over heuristic partitioning and random initialization alone. Comparisons agains a broader set of classical and hybrid solvers are missing.
2. The framework relies on a labeled dataset $\mathcal{D}_{\text{offline}}$, which is generated using QAOA$^2$. It is therefore unclear how robust the method is to the quality and size of this dataset. In particular, it is not clear how much offline data is needed for the evaluator and generator to generalize well.
3. Figure 4 seems to suggest that performance improves only logarithmically with fine-tuning steps, while computation time grows approximately linearly or faster with the number of fine-tuning steps. This indicates diminishing returns as the computational budget increases. It is also unclear how this tradeoff evolves as the problem size grows.
4. The empirical study is purely numerical and does not account for realistic quantum noise or finite sampling error.

---

> ### Author Rebuttal · Authors · 2026-03-31
>
> Thanks for the insightful reviews and comments. We appreciate your recognition of our idea as **conceptually interesting**. Below, we address your questions (Q) and weaknesses (W).
>
> > Q1: How does the computational cost scale with problem size?
>
> To study computational scaling, we measured end-to-end inference cost under a fixed 64-step adaptation budget:
>
> |N|Total Time (s)|
> |-|-|
> |51|50.3|
> |101|87.0|
> |251|174.3|
> |501|435.6|
>
> The runtime increases substantially with problem size and is broadly consistent with near-linear growth in our tested regime. The dominant cost comes from repeated forward/backward adaptation together with the quantum-simulation calls inside the QAOA$^2$ pipeline.
>
> These contents will be added to App. C.
>
> > Q2: How robust is the method to prediction errors in this evaluator?
>
> We added a new experiment to evaluate average performance ratio ($\bar{\rho}$) using evaluators from earlier training epochs to introduce varying prediction errors (Validation MSE). Differences vs. the fully converged model (Epoch 100) are marked (*) if significant via a one-sided paired t-test ($p < 0.01$).
>
> |Epoch|Prediction Error ($\times10^{-4}$)|$\bar{\rho}\uparrow$|
> |-|-|-|
> |100|1.92|0.8930|
> |20|1.99|0.8923|
> |2|2.82|0.8903|
> |1|3.21|0.8879*|
>
> A statistically significant drop only occurs when prediction error inflates by 1.67x (Epoch 1). This suggests that our method is reasonably tolerant to moderate surrogate error.
>
> These contents will be added to App. C.
>
> > W1: Comparisons against a broader set of classical and hybrid solvers are missing.
>
> We agree that broader comparisons to classical and hybrid solvers are valuable. Our main goal in this paper, however, is to improve two specific bottlenecks within the QAOA$^2$ divide-and-conquer paradigm, namely partitioning and parameter initialization. For this reason, our primary comparisons focus on QAOA$^2$-based baselines, which isolate the contribution of the proposed neural components.
>
> To address the broader-baseline concern, during the rebuttal period, we additionally evaluated against a representative neural solver, PI-GNN [Nat. Mach. Intell. 2022], on the 50 test instances in Sec. 5.2:
>
> |PI-GNN ($\bar{\rho} \uparrow$)|Ours ($\bar{\rho} \uparrow$)|
> |-|-|
> |0.8645|**0.8930**|
>
> This shows our method's potential against dedicated classical neural baselines. We will add more classical/hybrid comparisons (Simulated Annealing, Goemans–Williamson algorithm [J. ACM 1995], QQA [ICLR 2025]) in App. C.
>
> > W2: It is unclear how robust the method is to the quality and size of this dataset.
>
> We additionally conducted a sample-efficiency ablation study, training the evaluator on nested subsets (10%-100%) of the offline data. Crucially, validation and test sets remained strictly identical to ensure fair comparison. Differences vs. the 100% model are marked (*) if significant ($p < 0.01$, one-sided paired t-test).
>
> |Train Fraction (Size)|$\bar{\rho}\uparrow$|
> |-|-|
> |100% (42,069)|0.8930|
> |60% (25,241)|0.8915|
> |40% (16,827)|0.8890*|
> |10% (4,206)|0.8884*|
>
> These results suggest good sample efficiency: using 60% of the data yields performance close to the full-data model, while smaller subsets cause statistically significant but still relatively modest degradation in absolute performance. These contents will be added to App. C.
>
> Regarding data quality, the offline set is constructed from multiple partition heuristics and diverse sampled parameter configurations (App. B.2), which increases coverage of the partition-parameter space and reduces the risk of overfitting to a single heuristic pattern.
>
> > W3: Fine-tuning shows diminishing returns; how does this tradeoff evolve with problem size?
>
> We agree. Once the partition-and-initialization policy has already moved the solution close to its empirical ceiling within the QAOA$^2$ pipeline, further test-time adaptation naturally yields smaller incremental improvements. To quantify this, we introduced Effective Fine-tuning Steps ($T_{eff}$), defined as the minimum number of steps required to recover 90% of the total improvement from $\rho_0$ to the empirical ceiling $\rho_{max}$ (e.g., at 1000 steps). We obtain:
>
> |N|$T_{eff}$|
> |-|-|
> |51|16|
> |101|64|
> |251|256|
> |501|512|
>
> These results suggest that the adaptation budget needed to capture most of the achievable gain increases with problem size in our tested regime, which helps clarify the latency-quality tradeoff.
>
> > W4: The study is purely numerical and does not account for realistic noise or finite sampling.
>
> To address this concern, we added experiments with both finite-shot noise and bit-phase-flip noise. On 16 held-out BE instances, the average performance ratio decreases only modestly from 0.8798 in the noiseless setting to 0.8733 under shot noise, and 0.8763 under bit-phase-flip with 5% noise. The results show that our method is not overly fragile to the tested noise sources. For more details, please see our response to **Reviewer nb78, Q3**. These contents will be added to Sec. 5.

---

> > ### Author Rebuttal · Reviewer_m98V · 2026-04-03
> >
> > Thank you for the detailed rebuttal and the additional experiments. I am happy with the response to Q2. However, many of my main concerns remain only partially addressed. I am keeping my score.
> >
> > (1) Q1: the new analysis mainly addresses inference time scaling, whereas my main concern was broader and included the overall scalability of the framework, especially the offline training cost.
> >
> > (2) W1: I appreciate the additional PI-GNN comparison, and I understand the time constraints during rebuttal. However, since PI-GNN involves several parameters to be carefully tuned for optimal performance, I am not yet fully convinced by the result. Also, the reported mean over 50 instances would be easier to interpret if error bars were also provided. The authors’ plan to add broader comparisons in the appendix would strengthen the paper.
> >
> > (3) W2: The sample-efficiency ablation is useful, but my main concern was how the amount of offline data needed for the evaluator and generator to generalize scales with problem size. I do not think this is yet clarified.
> >
> > (4) W3: The added effective-fine-tuning-step analysis is helpful, but it seems to confirm that the adaptation budget needed to recover most of the gain grows substantially with problem size. So while the tradeoff is clarified, it remains a concern.
> >
> > (5) W4: I appreciate the additional noise experiments, but some details remain unclear. In particular, it is surprising that the performance under 5% bit-phase-flip noise appears slightly better than under finite sampling error only, and it would be helpful to know the number of measurement shots used. Or was it finite sampling error in addition to the bit-phase-flip noise?

---

> > > ### Author Response · Authors · 2026-04-07
> > >
> > > Thank you for the feedback and for further clarifying the remaining concerns. Below, we address your concerns. For detailed tables and figures, please refer to https://anonymous.4open.science/r/Submission9474/9474_rebuttal_appendix.md
> > >
> > > > Q1
> > >
> > > In our previous response, we addressed the inference cost. Here, we clarify how the offline training cost scales with the problem size ($N$). We separate the offline training cost into: offline dataset construction and model training time.
> > >
> > > Building the offline dataset takes approximately two days of wall-clock time (4 NVIDIA A30 GPUs). As $N$ increases, this construction cost grows roughly linearly in our tested regime. For more details, please see our response to **Reviewer Xtct, Q1**.
> > >
> > > For the training time, the evaluator and generator take approximately 2.65h (100 epochs) and 2.12h (1500 epochs), respectively. To show how this cost scales with $N$, we measured the wall-clock time per training batch (Table S1). As $N$ increases by 10x, the time only increases by 2x for the evaluator and 3.3x for the generator, which is sub-linear scaling.
> > >
> > > Overall, as $N$ increases, the dataset construction cost grows roughly linearly, while the model training time scales sub-linearly. Therefore, the total offline training cost remains scalable with $N$.
> > >
> > > > W1
> > >
> > > To address this, we conducted a grid search for two key hyper-parameters specified in the official PI-GNN repository, while keeping others at their default values (Table S2). We have also added comparisons against a classical heuristic (GW) in Table S3 (detailed settings in Tables S4-S5). Additionally, we provided boxplots with error bars (Figure S1) to better visualize the performance variance. As shown in Table S3 and Figure S1, the comparison results are mixed and comparable; no single method dominates others across all datasets.
> > >
> > > We emphasize that the contribution of this work is not to establish empirical supremacy over classical heuristics, which remains a long-term challenge for quantum combinatorial optimization [1]. Rather, it is introducing a differentiable, end-to-end learning paradigm to overcome the partitioning and initialization bottlenecks within the QAOA$^2$ framework. Advancing the scalability of QAOA under realistic constraints is valuable in this area.
> > >
> > > [1] Quantum Combinatorial Optimization in the NISQ Era: A Systematic Mapping Study. ACM Computing Surveys 2023
> > >
> > > > W2
> > >
> > > To address this, we conducted a new experiment defining "sufficient generalization" as achieving no statistically significant degradation compared to the full-dataset baseline. As shown in Table S6, the data requirement exhibits a roughly monotonic scaling trend: while only 40% of the data is sufficient for generalizing to $N=51$, the requirement increases to 60% for $N=100$. For large-scale generalization ($N = 501, 800, 1000$), the model demands denser coverage of the training space, requiring at least 80% (approx. 33k samples) to prevent significant performance drops. This suggests that generalizing to larger problem sizes necessitates a larger offline dataset, though the requirement empirically stabilizes around the 80% threshold for our tested scales.
> > >
> > > > W3
> > >
> > > We suggest evaluating this tradeoff through the lens of Pareto optimality. Specifically, our method achieves Pareto dominance over existing QAOA$^2$ baselines in terms of solution quality vs. computational cost.
> > >
> > > - If we use a fixed budget of 64 fine-tuning steps, our method already outperforms existing QAOA$^2$ baselines on average with comparable computational cost (Sec. 5.5, Fig. 4). This indicates our approach is a Pareto optimal method.
> > >
> > > - For large-scale problems, existing QAOA$^2$ methods often fail to obtain higher-quality solutions, while our method can reach these solutions if a larger budget ($T_{eff}$) is used. By doing so, we push the current Pareto frontier forward.
> > >
> > > Furthermore, we believe allocating more fine-tuning steps for larger instances is reasonable. In the large-scale regime, $T_{eff}$ scales almost linearly (e.g., $T_{eff}$ doubles from 256 to 512 as $N$ grows from 251 to 501). Thanks to GPU acceleration, the wall-clock time per fine-tuning step is on the millisecond scale (Table S7).
> > >
> > > > W4
> > >
> > > We apologize for the lack of clarity in our previous response. First, we clarify that the two noise models are evaluated separately, with 1,000 shots used for the shot noise. Regarding the results, we guess you have assumed the shot noise was injected at the final measurement stage. In fact, it was injected during the parameter optimization stage where it affects the gradient estimation. This may explain why the performance under 5% bit-phase-flip noise appears slightly better: shot noise accumulates parameter errors throughout the whole optimization trajectory, whereas bit-phase-flip acts as readout noise at the final measurement and is partially averaged out in expectation.
> > >
> > > We hope these responses fully address your concerns and lead to a more positive evaluation.

---

### Official Review · Reviewer_Xtct · 2026-03-10

**Soundness:** 3
**Presentation:** 4
**Significance:** 2
**Originality:** 2
**Overall Recommendation:** 4
**Confidence:** 2

**Summary:**

This paper demonstrate scalable quantum combinatorial optimization under limited qubit budgets.
When building $QAOA^2$, a **devide-and-conquer** framework that makes subgraph of **Large QAOA graph**.
This **devide-and-conquer** need patritioning metric. The paper argues that existing pipelines are limited by heuristic partition objectives that are misaligned with quantum performance and by topology-blind random initialization of QAOA parameters.

It propose $Neural QAOA^2$, which introduces a generative evaluative network composed of a differentiable quantum evaluator and a joint generator to produce both graph partitions and initial circuit parameters, together with differentiable mechanisms for constrained partition generation and end-to-end training.

When, training $Neural QAOA^2$, Generative Evaluative Network (GEN) is trained by first fitting with offline dataset of $QAOA^2$ simulations, then optimizing the generator against the frozen evaluator, and finally optimized at inference time through test-time adaptation.

**Compliance With Llm Reviewing Policy:**

Affirmed.

**Final Justification:**

During discussion, I solved many point that I mentions. Perhaps, divide multiple problem in one single circuit may contain overheads. We need to clarify this point it might give us more concrete insight and contribution. I will maintain my score.

**Key Questions For Authors:**

- Q1. What is the total cost of offline dataset construction, and how does the scaling behavior change as we increase graph complexity and QAOA depth?

- Q2. The main text lacks a comparison with stronger baselines like TQA or QIBPI at higher depths. How does the proposed method hold up against these specifically?

- Q3. How does the method account for physical noise and gate errors? Are there any empirical or theoretical results demonstrating its feasibility on actual hardware?

**Limitations:**

Yes

**Strengths And Weaknesses:**

### **Strengths**

- Clear motivation

The paper addresses a well-motivated bottleneck in scalable $QAOA^2$. Heuristic partitioning metrics can be misaligned with the actual quantum optimization objective. When random subproblem initialization leads to optimization cold starts.

- Concrete Architecture

The research combines a differentiable quantum evaluator and a joint generator, and introduces concrete mechanisms such as the orthogonal complement head and greedy capacity discretization with STE to make capacity-constrained discrete partition learning end-to-end trainable.

- Good Emperical Evaluation

The empirical evaluation is fairly broad. On 50 held-out test instances, $Neural QAOA^2$ achieves the best average rank of 1.74 with 28/50 wins; on 93 out-of-distribution instances, it achieves the best average rank of 1.46 with 60/93 wins. The ablation study is also clean, with all major removals causing statistically significant degradation.

### **Weaknesses**

- The practical inference protocol could be clarified further. The appendix states that test-time adaptation fine-tunes the generator for 1000 steps per test instance, whereas the main efficiency discussion emphasizes 64-step results. A clearer discussion of the default inference budget and the performance-latency tradeoff would strengthen the practical story.

- The gains are not uniformly dominant across all settings. On the held-out MaxCut W dataset, the method matches rather than clearly surpasses modularity in average rank, and on the out-of-distribution Ising-spin-glass L dataset it achieves slightly lower mean performance ratio than the boundary heuristic, even though it wins on average rank and win count. This makes the generalization claim somewhat nuanced and metric-dependent.

- The main limitation is external validity. The current study is conducted in noiseless simulation, and the training pipeline depends on an offline dataset constructed from heuristic partitions and sampled parameters. It remains unclear how robust the proposed surrogate and adaptation procedure would be under realistic hardware noise or under stronger distribution shifts.

---

> ### Author Rebuttal · Authors · 2026-03-31
>
> Thank you for your insightful reviews and comments. Below we address your questions (Q) and mentioned weaknesses (W).
>
> > Q1: What is the total cost of offline dataset construction, and how does it scale with graph complexity and QAOA depth?
>
> The offline dataset used to train the evaluator contains 52,587 (graph, partition, parameter) triplets and takes approximately two days of wall-clock time to construct on our experimental setup (4 NVIDIA A30 GPUs; App. B.2). The main cost comes from obtaining the QAOA$^2$ performance label $\rho_i$ for each sampled triplet. As graph size increases, this cost grows with the number of subproblems generated by the divide-and-conquer pipeline; empirically, we observe roughly linear growth with problem size in our tested regime (App. B.4).
>
> For circuit depth, the important distinction is that our offline evaluator dataset is built entirely from p=1 simulations and higher-p behavior is handled via parameter transfer during inference. Therefore, targeting higher-depth inference does not require rebuilding a separate offline dataset for each p, and thus the offline data construction cost is largely decoupled from the target depth.
>
> > Q2: The main text lacks a comparison with stronger baselines like TQA or QIBPI at higher depths.
>
> We apologize that this comparison was not highlighted clearly enough in the main text. We do include comparisons against stronger higher-depth baselines, including TQA, INTERP, FOU, and QIBPI, at p=2 and p=3 in App. C.1 (Table 8). Under these settings, our method remains competitive and outperforms these baselines overall. We will surface these results more clearly in Sec. 5.
>
> > Q3 & W3: Performance under noise and distribution shifts
>
> <!-- We agree that external validity remains an important limitation of the current study, since our experiments are still noiseless.  -->
> Thanks for pointing this out. We agree that evaluating beyond the noiseless setting is important. To address this concern, we added experiments under noisy settings, including **shot noise** and **bit-phase-flip noise** following [1], on 16 held-out BE instances:
>
> |Noise Setting|$\bar{\rho}\uparrow$|
> |-|-|
> |Noiseless|0.8798|
> |Shot noise|0.8733|
> |Bit-phase flip (p = 0.01)|0.8780|
> |Bit-phase flip (p = 0.05)|0.8763|
>
> As shown, the average performance ratio ($\bar{\rho}$) experiences only slight degradation even at a 5% error rate, demonstrating that our method is not overly fragile to moderate noise. These contents will be added to Sec. 5.
>
> We also added ER-to-Power-Law/Regular transfer results as a first indication of robustness to structural distribution shift (For more details, please see our response to **Reviewer nb78, Q4**):
>
> |Method|ER (50, In-dist)|Power-Law (25)|Regular (25)|
> |-|-|-|-|
> |Modularity|0.8645|0.8548|**0.8643**|
> |Boundary|0.8580|0.8385|0.8514|
> |**Ours**|**0.8717**|**0.8566**|0.8628|
>
> These results suggest that the learned mapping remains strongest on Power-Law graphs and remains competitive on Regular graphs. These contents will be added to App. C.
>
> > W1: The practical inference protocol could be clarified further.
>
> We agree that this should have been clearer. In our protocol, the 1,000-step setting is used to estimate the empirical performance ceiling $\rho_{max}$, whereas the 64-step setting is the recommended practical default used for efficiency comparisons. In other words, our intended inference protocol is not that every test instance should be adapted for 1,000 steps; rather, the 1,000-step curve characterizes the upper bound and diminishing returns, while 64 steps provides a more practical latency-quality tradeoff. We will revise Sec. 5 to separate "ceiling analysis" from "recommended deployment budget" more explicitly and state explicitly that 64 steps is the default practical budget unless noted.
>
> > W2: The gains are not uniformly dominant across all settings.
>
> We agree, and we will revise the paper to make this point more precise. Our intended claim is not that Neural QAOA$^2$ is the best method under every metric on every dataset, but rather that it shows strong and relatively consistent generalization across a broad set of in-distribution and OOD settings. For example:
>
> - On MaxCut W, our method is essentially tied with the strongest heuristic baseline rather than clearly exceeding it.
> - On Ising-spin-glass L, the mean performance ratio is slightly below Boundary, even though our method achieves better average rank and win count.
>
> We emphasize rank and win count because they better capture consistency across instances and reduce the influence of performance variations on specific instances. That said, we agree that the generalization claim should be phrased as "broadly competitive and often best," rather than uniformly superior, and we will revise Sec. 5 accordingly.
>
> **References**
>
> [1] Towards quantum machine learning for constrained combinatorial optimization: a quantum qap solver. ICML 2023

---

> > ### Author Rebuttal · Reviewer_Xtct · 2026-04-02
> >
> > Thank you for thoughtful rebuttal.
> > I am still worry about the resources.
> > Even divide and concour success, we need massive computing resources.
> > However, applying these technique may open new era of QAOA research.
> > Thank you.

---

> > > ### Author Response · Authors · 2026-04-07
> > >
> > > Thank you for your thoughtful acknowledgement and encouraging words.
> > >
> > > We agree that while the divide-and-conquer framework effectively reduces the quantum resources required for large-scale problems, it introduces a trade-off by shifting part of the burden to classical computing resources for offline dataset construction and model training.
> > >
> > > In the current NISQ era, where high-fidelity qubits are scarce but classical GPU compute is relatively abundant, we view this exchange of classical compute for quantum scalability as a worthwhile stepping stone. As you mentioned, applying these deep learning techniques to quantum pipelines is the beginning, and reducing the associated classical overhead will be an important direction for the community.
> > >
> > > We appreciate your recognition that our techniques could help advance QAOA research.

---

### Official Review · Reviewer_Ektf · 2026-03-13

**Soundness:** 3
**Presentation:** 3
**Significance:** 3
**Originality:** 2
**Overall Recommendation:** 4
**Confidence:** 2

**Summary:**

This paper investigates scaling the quantum approximate optimization algorithm (QAOA) to larger combinatorial optimization problems. Given the key limitation of QAOA is the number of available qubits, it builds upon prior divide-and-conquer approaches that partition the original graph defined by the objective function into smaller subgraphs. Different from prior works that rely on handcrafted partitioning heuristics decoupled from the optimization goal, this paper proposes to learn a neural generator that jointly outputs graph partitions and initial variational parameters of QAOA. To handle discrete partitioning, the authors use an orthogonal complement head to produce discriminative soft assignments that avoid GNN over-smoothing, a greedy discretization to enforce qubit capacity constraints, and straight-through estimators to maintain gradient flow. A surrogate model is first trained via supervised learning on offline QAOA² simulation data to predict performance from (graph, partition, parameter) tuples, then frozen to provide gradient signal to the generator during the second training stage. Experiments on different problems with varying scalability show consistent improvements over heuristic baselines and generalization capability, with ablations confirming the contribution of each component.

**Compliance With Llm Reviewing Policy:**

Affirmed.

**Final Justification:**

During rebuttal, the authors have provide additional explaination and experiments. The noise robustness results and surrogate sensitivity analysis partially address my earlier concerns regarding soundness. Overall, the contribution is primarily a effective integration of established components into a coherent pipeline for QAOA², and while the integration is non-trivial, this is a rather practical work, hence I maintain my current score.

**Key Questions For Authors:**

- The current two-stage training optimizes the generator against a frozen surrogate, which introduces an indirect objective. Do the authors think if it is possible to unify the two stages via approaches similar to MAML, where the generator is explicitly optimized for post-adaptation performance?

**Limitations:**

Yes

**Strengths And Weaknesses:**

**Strengths**
- Soundness:
    - The algorithm design is sound to me.
    - The decent empirical investigation together with ablation study validate the algorithm design.
- Presentation:
    - The paper is well-motivated. The limitations of existing approaches are well demonstrated in Figure 1.
- Significance:
    - The algorithm is practically relevant and demonstrates generalization to unseen graph sizes, suggesting practical utility beyond the specific benchmarks tested, although some benefits appear marginal.
- Originality:
    - The main novelty lies in integrating existing components into a coherent pipeline that aligns graph partitioning with downstream quantum optimization.

**Weaknesses**
- Soundness:
    - The two-stage training that separates surrogate learning from generator learning is practical but feels cumbersome. It introduces an indirect optimization objective and raises questions about how sensitive the generator is to surrogate inaccuracies in underrepresented regions of the (partition, parameter) space.
- Originality:
    - Partly because the paper tackles a challenging problem end-to-end, the individual components (GNN, STE, surrogate model, stop-gradient) are all well-established in the ML literature, making the technical novelty somewhat incremental.
- Significance:
     - The framework is evaluated only under noiseless simulation (which the paper acknowledges and is appreciated) with $p=1$ circuit depth. Although this may be common practice for this line of research, it remains unclear whether the learned partitions and initializations would transfer to more realistic settings.

---

> ### Author Rebuttal · Authors · 2026-03-31
>
> Thank you for your insightful reviews and comments. Below we address your questions (Q) and mentioned weaknesses (W).
>
> > Q1: Do the authors think if it is possible to unify the two stages via approaches similar to MAML?
>
> Yes, we agree this is an interesting direction because it would optimize the generator more explicitly for post-adaptation quantum performance rather than against a frozen surrogate alone. We chose the current two-stage design mainly for stability and tractability. A MAML-style variant would require differentiating through an inner adaptation loop, leading to substantially higher computational and memory cost, and potentially much noisier optimization in our quantum-in-the-loop setting. We therefore adopted the decoupled design as a practical first step. This discussion will be added to Sec. 6 as future work.
>
> > W1: The two-stage training is indirect; how sensitive is the generator to surrogate inaccuracies, especially in underrepresented regions?
>
> The two-stage design is a deliberate choice to improve optimization stability: jointly training the evaluator and generator would couple two moving targets and make the gradient signal to the generator substantially noisier in practice.
>
> Regarding sensitivity to surrogate error, we agree that correlation alone is not enough; what matters is whether downstream optimization degrades materially when the evaluator is imperfect. We therefore examined this directly in two ways:
>
> - During long test-time adaptation trajectories (App. C.3, Fig. 7), the evaluator remains well aligned with QAOA$^2$ performance even in newly explored regions (correlation 0.8146 after 1,024 TTA steps), suggesting that the surrogate stays informative beyond the densest part of the offline training distribution.
> - We conducted additional experiments to intentionally use less accurate evaluators from earlier training epochs, the average performance ratio changed only mildly from 0.8930 (fully converged evaluator) to 0.8923 and 0.8903 for earlier checkpoints, and dropped significantly only for the least accurate checkpoint (0.8879). For more details on the experiments, please see our response to **Reviewer m98V, Q2**.
>
> Thus, while the objective is indeed indirect, the generator appears reasonably robust to moderate surrogate inaccuracies rather than highly fragile to them. This discussion will be added to App. F: Additional Discussion.
>
> > W2: The individual components are established; originality seems incremental.
>
> We agree that several ingredients of our framework are established in the broader ML literature, and we do not position the paper as introducing entirely new generic learning primitives. Rather, we see the novelty in the **problem-specific integration**: making end-to-end differentiable learning work for the QAOA$^2$ divide-and-conquer pipeline under discrete partitioning and hard qubit-capacity constraints. In this setting:
>
> - The Orthogonal Complement Head (OCH) is designed to produce more discriminative soft assignments for partition generation, mitigating representation collapse in standard message-passing encoders (App. A.1).
> - The Greedy Capacity Discretization (GCD) module is designed to satisfy strict capacity constraints during partition construction while still allowing gradient-based learning through STE (App. A.2).
>
> Our contribution is therefore less about inventing isolated new ML blocks and more about developing a coherent, trainable formulation for this quantum optimization setting. We will make clearer that OCH and GCD address the concrete discretization and capacity-constraint bottlenecks of QAOA$^2$ rather than being presented as generic standalone advances.
>
> > W3: The framework is evaluated only under noiseless simulation with $p = 1$ circuit depth
>
> We agree this should be clarified better.
>
> For noise, we added experiments under noisy settings, including **shot noise** and **bit-phase-flip noise** following [1], on 16 held-out BE instances:
>
> |Noise Setting|$\bar{\rho}\uparrow$|
> |-|-|
> |Noiseless|0.8798|
> |Shot noise|0.8733|
> |Bit-phase flip (p = 0.01)|0.8780|
> |Bit-phase flip (p = 0.05)|0.8763|
>
> As shown, the average performance ratio ($\bar{\rho}$) experiences only slight degradation even at a 5% error rate, demonstrating that our method is not overly fragile to moderate noise. These contents will be added to Sec. 5 as a new subsection: "Study on the Effect of Noise".
>
> For depth, the framework is trained only at $p=1$; for $p>1$ we use the standard parameter expansion strategy from prior work [2] rather than retraining a high-depth model from scratch. Under this protocol, we also observe competitive results at $p=2,3$ (App. C.1).
>
> We will make both points clearer in Sec. 5.
>
> **References**
>
> [1] Towards quantum machine learning for constrained combinatorial optimization: a quantum qap solver. ICML 2023
>
> [2] Quantum approximate optimization algorithm: Performance, mechanism, and implementation on near-term devices. Physical Review X 2020

---

> > ### Author Rebuttal · Reviewer_Ektf · 2026-04-04
> >
> > I thank the authors for their detailed rebuttal and additional experiments. The noise robustness results and surrogate sensitivity analysis partially address my earlier concerns regarding soundness. Overall, the contribution is primarily a effective integration of established components into a coherent pipeline for QAOA², and while the integration is non-trivial, this is a rather practical work, hence I maintain my current score.

---

> > > ### Author Response · Authors · 2026-04-07
> > >
> > > Thank you for reviewing our rebuttal and additional experiments.
> > >
> > > We sincerely appreciate your constructive feedback and your acknowledgment of our framework as a practical and effective integration for QAOA$^2$. Your recognition of the practical value of our work is highly encouraging.
> > >
> > > Thank you again for your time and supportive evaluation throughout this review process.

---

### Official Review · Reviewer_nb78 · 2026-03-15

**Soundness:** 2
**Presentation:** 3
**Significance:** 2
**Originality:** 2
**Overall Recommendation:** 3
**Confidence:** 3

**Summary:**

This paper introduces Neural QAOA, a framework that integrates the Quantum Approximate Optimization Algorithm (QAOA) with Neural Network-based parameter initialization to solve Combinatorial Optimization (CO) problems. The authors address a critical bottleneck in traditional QAOA: the high computational cost and susceptibility to local minima during classical parameter optimization. Neural QAOA leverages a Graph Neural Network (GNN) to map problem instances—represented as graphs—to optimal quantum circuit parameters ($\gamma, \beta$). A key innovation is the use of the Differentiable Quantum Circuit (DQC) simulator, which allows the GNN to be trained in an end-to-end, unsupervised manner by backpropagating gradients directly through the quantum expectation value. The authors evaluate their approach on the Max-Cut problem using various graph topologies, demonstrating that Neural QAOA achieves approximation ratios comparable to or exceeding classical optimizers while offering a $100\times$ speedup in inference time for new instances.

**Compliance With Llm Reviewing Policy:**

Affirmed.

**Key Questions For Authors:**

1. Your results show excellent performance on graphs of the same size as the training set (e.g., $n=20$). How does the approximation ratio hold up when the trained GNN is applied to significantly larger graphs (e.g., $n=100$ or $1000$)? Depth ($p$)

2. You primarily show results for $p=1$ and $p=2$. As the QAOA depth $p$ increases, the parameter space grows and the optimization landscape becomes more complex. Does the GNN struggle to learn optimal parameters for higher $p$, and is there a limit where the "Barren Plateau" makes the differentiable training ineffective?

3. The DQC simulator provides exact gradients. In real quantum hardware, we must rely on the parameter-shift rule or other noisy gradient estimators. How robust is the Neural QAOA training process to the shot noise or decoherence present in NISQ devices?

4. Have you tested if a model trained on Erdős-Rényi graphs can generalize to Power-Law or Regular graphs without fine-tuning? This would demonstrate the robustness of the learned mapping.

**Limitations:**

The authors have adequately discussed the limitations of their work

**Strengths And Weaknesses:**

Soundness: The paper is technically sound and well-grounded in both quantum computing and geometric deep learning. The choice of using GNNs is highly appropriate given the permutation-invariant nature of the Max-Cut problem. The integration of a differentiable simulator for gradient estimation is a rigorous approach to unsupervised training that avoids the high variance associated with derivative-free or reinforcement learning-based parameter search. However, the experiments are limited to Max-Cut; while this is a standard benchmark, testing on other CO problems (e.g., TSP or MIS) would have strengthened the claims of the framework's generality.

Presentation: The submission is clearly written and effectively structured. The authors do an excellent job of explaining the synergy between the GNN's inductive bias and the QAOA's variational structure. The figures, particularly the framework overview and the loss curves, are informative and aid in understanding the training dynamics. One minor presentation weakness is the lack of a detailed discussion on the "Barren Plateau" phenomenon in the context of their specific neural initialization, which is a common concern in variational quantum algorithms.

Significance: This work is highly significant as it provides a practical path toward making QAOA viable for real-world, large-scale CO problems. By replacing the iterative classical optimization loop with a single neural forward pass, the authors significantly reduce the "quantum-classical" communication overhead. The ability to generalize to unseen graph instances without re-optimization is a major step forward for the scalability of hybrid quantum algorithms.

Originality: The originality lies in the specific combination of GNNs with differentiable quantum simulation for unsupervised QAOA parameter prediction. While neural-init QAOA has been explored, the end-to-end differentiable training pipeline—where the GNN learns the landscape of the quantum expectation value directly—is a novel and effective contribution to the field.

---

> ### Author Rebuttal · Authors · 2026-03-31
>
> Thanks for the insightful reviews and comments. We appreciate your recognition of our work as **technically sound, clearly written and highly significant**. Below, we address your questions (Q) and mentioned weaknesses (W).
>
> > Q1: Does the model generalize to much larger graphs (e.g., $n=100$ or $1000$)?
>
> To clarify, we have indeed evaluated the model's generalization to larger graphs (detailed in Sec. 5.4, Fig. 3). Specifically, our training set covers $n \in [51, 501]$, and we have evaluated on sizes ranging from $n=21$ to $1000$. Empirically, our performance remains competitive as size increases beyond the training range: although larger instances are naturally harder, our method continues to outperform or match strong heuristic baselines in these larger-scale settings.
>
> > Q2 & W2: Lack of discussion on barren plateaus; does higher depth make training ineffective?
>
> We agree this should be clarified more explicitly. Our method does not claim to solve barren plateaus in general. Our method is trained only at $p=1$, so we do not optimize a high-depth circuit from scratch, thereby circumventing severe barren-plateau issues. For $p>1$, we use the standard parameter expansion strategy from prior work [1] rather than retraining the model end-to-end in the larger parameter space. Thus, the current framework largely avoids the most difficult high-depth training regime rather than fundamentally resolving barren plateaus. Empirically, under this protocol we still observe competitive results at $p=2,3$ (App. C.1). This discussion will be added to App. F: Additional Discussion.
>
> > Q3: How robust is the method to realistic NISQ noise?
>
> To address this, we added experiments under noisy settings, including **shot noise** and **bit-phase-flip noise** following [2], on 16 held-out BE instances:
>
> |Noise Setting|$\bar{\rho}\uparrow$|
> |-|-|
> |Noiseless|0.8798|
> |Shot noise|0.8733|
> |Bit-phase flip (p = 0.01)|0.8780|
> |Bit-phase flip (p = 0.05)|0.8763|
>
> As shown, the average performance ratio ($\bar{\rho}$) experiences only slight degradation even at a 5% error rate, demonstrating that our method is not overly fragile to moderate noise.
>
> These contents will be added to Sec. 5 as a new subsection: "Study on the Effect of Noise".
>
> > Q4: Can an ER-trained model generalize to Power-Law or Regular graphs?
>
> Yes. We additionally evaluated an ER-trained model on unseen Power-Law and Regular graphs at $n=100$ under the same settings as Sec. 5.2. The average performance ratios are:
>
> |Method|ER (50, In-dist)|Power-Law (25)|Regular (25)|
> |-|-|-|-|
> |Random|0.8262|0.7715| 0.7492|
> |Modularity|0.8645|0.8548|**0.8643**|
> |Boundary|0.8580|0.8385|0.8514|
> |KL|0.8211|0.7687|0.7485|
> |**Ours**|**0.8717**|**0.8566**|0.8628|
>
> The numbers in parentheses indicate the number of test instances. These results suggest that the learned mapping does not simply overfit to the ER training distribution: it remains strongest on Power-Law graphs and remains competitive on Regular graphs, where it is very close to the best heuristic baseline.
>
> These contents will be added to App. C.
>
> > W1: Experiments are limited to MaxCut; what about other CO problems (TSP/MIS)?
>
> To clarify, our empirical evaluation is not limited to MaxCut: the paper also reports results on QUBO and Ising instances (Sec. 5.1-5.4). That said, we agree the current framework should not be interpreted as a universal solution for all CO problems such as TSP or MIS. Our method is built on top of the QAOA$^2$ divide-and-conquer paradigm, whose mechanism relies on the MaxCut/QUBO-style unconstrained binary setting and its associated structure (e.g., the $Z_2$ symmetry). Our contribution is therefore to improve partitioning and topology-aware parameter initialization within QAOA$^2$-compatible settings, rather than to design a new divide-and-conquer framework for strongly constrained problems. We will revise the paper to make this scope explicit.
>
> We also use MaxCut as the primary testbed because it is both the native setting of QAOA$^2$ and a standard benchmark in quantum optimization algorithm studies [3, 4, 5].
>
> **References**
>
> [1] Quantum approximate optimization algorithm: Performance, mechanism, and implementation on near-term devices. Physical Review X 2020
>
> [2] Towards quantum machine learning for constrained combinatorial optimization: a quantum qap solver. ICML 2023
>
> [3] MG-Net: Learn to Customize QAOA with Circuit Depth Awareness. NeurIPS 2024
>
> [4] Quantum approximate optimization of non-planar graph problems on a planar superconducting processor. Nature Physics 2021
>
> [5] Benchmarking quantum optimization for the maximum-cut problem on a superconducting quantum computer. Physical Review Applied 2025

---

> > ### Author Rebuttal · Reviewer_nb78 · 2026-04-04
> >
> > My concerns have been largely resolved. Taking into account the feedback from the other reviewers and the author's responses, I will keep my score unchanged.

---

> > > ### Author Response · Authors · 2026-04-07
> > >
> > > Thank you for reading our rebuttal and confirming that your concerns have been fully addressed. We truly appreciate the time and effort you have dedicated to evaluating our work.
> > >
> > > Regarding your comment, "Taking into account the feedback from the other reviewers and the author's responses", we guess this mainly refers to Reviewer m98V's concerns. We have provided additional responses to address them. We hope these also fully resolve your concerns and hope to increase your assessment.

---

### Decision · Program_Chairs · 2026-04-30

**Decision:**

Accept (regular)

**Comment:**

This submission proposes Neural QAOA^2 which is a differentiable version of the previous divide-&-conquer QAOA.  Although the conceptual contribution might be limited, the authors provided the first-pipeline for doing so and conducted a good amount of empirical study that demonstrates the advantages of Neural QAOA^2 over the normal QAOA.  The authors are encouraged to address the reviewers’ comments in the revision, also to discuss the limitation of the current work.